# Raman Spectroscopy and Imaging Studies of Human Digestive Tract Cells and Tissues—Impact of Vitamin C and E Supplementation

**DOI:** 10.3390/molecules28010137

**Published:** 2022-12-24

**Authors:** Karolina Beton-Mysur, Beata Brozek-Pluska

**Affiliations:** Laboratory of Laser Molecular Spectroscopy, Institute of Applied Radiation Chemistry, Lodz University of Technology, Wroblewskiego 15, 93-590 Lodz, Poland

**Keywords:** Raman imaging, Raman spectroscopy, cell culture, cancer, antioxidants, supplementation

## Abstract

Cancers of digestive tract such as colorectal cancer (CRC) and gastric cancer (GC) are the most commonly detected types of cancer worldwide and their origin can be associated with oxidative stress conditions. Commonly known and followed antioxidants, such as vitamin C and E, are widely considered as potential anti-cancer agents. Raman spectra have great potential in the biochemical characterization of matter based on the fact that each molecule has its own unique vibrational properties. Raman spectroscopy allows to precisely characterize components (proteins, lipids, nucleic acids). The paper presents the application of the Raman spectroscopy technique for the analysis of tissue samples and cells of the human colon and stomach. The main goal of this study is to show the differences between healthy and cancerous tissues from the human digestive tract and human normal and cancer colon and gastric cell lines. The paper presents the spectroscopic characterization of normal colon cells, CCD-18 Co, in physiological and oxidative conditions and effect of oxidative injury of normal colon cells upon supplementation with vitamin C at various concentrations based on Raman spectra. The obtained results were related to the Raman spectra recorded for human colon cancer cells—CaCo-2. In addition, the effect of the antioxidant in the form of vitamin E on gastric cancer cells, HTB-135, is presented and compared with normal gastric cells—CRL-7869. All measured gastric samples were biochemically and structurally characterized by means of Raman spectroscopy and imaging. Statistically assisted analysis has shown that normal, ROS injured and cancerous human gastrointestinal cells can be distinguished based on their unique vibrational properties. ANOVA tests, PCA (Principal Component Analysis) and PLSDA (Partial Least Squares Discriminant Analysis) have confirmed the main role of nucleic acids, proteins and lipids in differentiation of human colon and stomach normal and cancer tissues and cells. The conducted research based on Raman spectra proved that antioxidants in the form of vitamin C and E exhibit anti-cancer properties. In consequence, conducted studies proved that label-free Raman spectroscopy may play an important role in clinical diagnostic differentiation of human normal and cancerous gastrointestinal tissues and may be a source of intraoperative information supporting histopathological analysis.

## 1. Introduction

Gastrointestinal tract cancer is a term for a group of all cancers localized within the digestive system and intestinal tract, which includes colorectal cancer (CRC) and gastric cancer (GC). All cancers constitute a serious public health problem worldwide; however, those related to the digestive system are some of the more difficult to diagnose due to their painless development [1].

CRC is the third most common cancer in men and the second in women worldwide, and it can be characterized with high metastasis and poor prognosis [2,3,4]. Development of CRC is correlated with aging, diet, lifestyle (smoking, alcohol consumption, physical activity) and genetic load [5,6]. Colon cancer develops from adenomatous polyps that can progress into cancer or metastatic colon cancer [7]. Strategies to prevent the occurrence and development of CRC comprise implementation of Fecal Occult Blood (FOB) tests, endoscopic reconnaissance of high-risk individuals for detection and removal of malignant lesions [6,8].

The majority of stomach cancer cases (approx. 95 percent malignant tumors) are adenocarcinomas, i.e., tumors that originate from the cells of the glandular epithelium of the stomach. The remaining 5 percent are lymphomas, sarcomas, carcinoids or stromal tumors. Clinically, these cancers are classified according to the Lauren’s classification into two types: diffuse and intestinal [9]. This is an important division, as these groups differ in prognosis, occurrence and the basis for the development of the disease. The main reason of gastric cancer development is infection by *Helicobacter pylori*, although a style of life and genetic load are also important factors [10,11]. The conventional diagnostic techniques for gastric cancer screening are incisional/excisional biopsy, upper endoscopy and contrast radiography [12,13,14].

Application of medications is another strategy in successful anti-cancer therapy [6,15]. Despite the continuous development and research into synthetic anticancer drugs, there is still a need to discover and research the most natural compounds, especially those available in nature, in terms of their chemotherapeutic and chemopreventive activity against the onset and progression of cancer [16] Many molecules derived from plants and microorganisms, such as alkaloids, statins, polyphenols, isoprenoids and vitamins, can be considered in the treatment or prevention of cancer [17,18]. Particular attention should be paid to vitamin A, C and E due to their numerous beneficial properties [19].

Vitamins are organic molecules involved in metabolic processes and cellular regulations, and are indispensable for growth and development [20] Vitamin C is a vitamin that has sparked much debate with regard to the prevention of many diseases such as atherosclerosis, neurodegenerative disorders, diabetes and cancer, where it would act as a therapeutic agent [21] Vitamin C, also known as L-ascorbic acid (AA), is a powerful antioxidant and free radical scavenger [19] Unfortunately, through evolution, the human body has lost the ability to synthesize vitamin C endogenously, so its supplementation can only take place through its absorption in the diet. It occurs naturally in fruits and vegetables [21] but it can also be produced industrially [22] Vitamin C is also a therapeutic agent for many diseases such as scurvy and colds, and is considered a therapeutic substance in the treatment of various types of cancer [21].

Vitamin C has been found to have an antiproliferative effect on melanoma [23] gastric adenocarcinoma [24] breast cancer [25] pancreatic cancer [26] and also colon cancer cells [27] Although AA is a widely known antioxidant, it is said to react at high concentrations as a pro-oxidant, and its anti-cancer effect is attributed to its ability to induce self-oxidation [23,25]. Hence, elevated levels of reactive oxygen species (ROS) due to the hydrogen peroxide formed cause cancer cell toxicity. Induction of apoptosis [27] autophagy [26] mitochondrial structure impairment [28] and mitochondrial Ca^2+^ release have been found to be a consequence of increased ROS production after exposure to high doses of vitamin C [29].

Vitamin E exists in the form of eight congeners: four tocopherols with a saturated side chain and four analogous tocotrienols with three double bonds in the side chain. In both groups there are four forms: α, β, γ and δ, differing in the number of methyl substituents on the phenyl ring. Each of the eight forms of vitamin E exhibits slightly different biological activity. In the human body, α-tocopherol plays the most important role. It is the main antioxidant found in cells (for unsaturated fatty acids and vitamin A). It conditions the proper structure of biological membranes and protects them. It enables the synthesis of some lipids, as well as influences muscle metabolism. It prevents cardiovascular diseases and affects blood clotting. Vitamin E reduces the likelihood of cancer development by destroying free radicals that damage the DNA of the body’s cells. Vitamin E is lipid—Soluble compound that is localizes in lipid regions of cell membranes. Vitamin E has antioxidant properties obtained by transferring the hydrogen ion from the phenolic group on the chromanol ring, thus preventing lipid peroxidation harmful to the body. In this way, it maintains the integrity of the cell membrane and acts as a specific lipid scavenger of superoxide radicals [30,31]. The natural sources of vitamin E are vegetables, fruits and nuts. α-tocopherol can be found in products such as peanut, spinach, sunflower, cottonseed and safflower oil, and γ-tocopherol is predominantly found in corn, linseed, tomatoes, beet leaves, dried apricots, soybean and rapeseed oil [32].

Tocopherols are considered to be potential group of compounds for the prevention and treatment of atherosclerosis [30] Vitamin E in different forms showed also anti-proliferative activity against various cancer cell lines [33,34,35,36,37]. As a consequence of exposure to vitamin E interference in de novo sphingolipids synthesis [33] vacuole formation and chromatin condensation [34,37], reduction in DNA synthesis and fragmentation of DNA and apoptosis induction [36,37] was reported.

Tocopherols are mainly associated with antioxidation because they inactivate free radicals and prevent oxidative stress. They protect against peroxidation of polyunsaturated fatty acids which are part of phospholipids of cell membranes and blood plasma lipoproteins. They also regulate the activity of enzymes. One of them is protein kinase C, which is involved in the transmission of signals within cells. α-tocopherol may affect the activity of enzymes associated with the formation of inflammation [38].

Diagnostic techniques for detection of CRC and GC possess their own specific advantages and disadvantages in terms of their accessibility and accuracy of diagnostic outcomes [8,14]. Therefore, it is necessary to investigate new techniques enabling rapid, accurate and unambiguous detection of cancer presence and development in the gastrointestinal tract tissues. Over the past hundred years, a lot of attention has been paid to developing and running a check of spectroscopic techniques, such as Infrared Spectroscopy and Raman spectroscopy (RS), especially in context of their application for medical diagnosis. RS is a non-destructive spectroscopic technique the aim of which is to analyze inelastic scattering of light by vibrating molecules and to obtain information about the chemistry and structure of the investigated molecule/sample. The most often used exciting laser beams in this technique are 355 nm (blue), 532 nm (green), 633 nm (red) or 785 nm (red) [39,40].

RS was reported to apply for detection of skin diseases (atopic eczema, allergy, melasma) [41] but also for different types of cancer (breast, lung, brain, pancreatic, prostate, ovarian, gastrointestinal) [40,42] with successful differentiation between cancer and healthy tissues. Indeed, Raman spectroscopy and imaging can be also a useful tool for the structural characterization of the single cells, including brain [43] breast [44] and even colon [45,46], as well as the whole tissue areas.

Considering all above-mentioned information, occurrence of CRC and GC is a consequence of redundant oxidative stress generation, which causes DNA damage which finally leads to carcinogenesis [3,47]. Vitamin C and E were reported to cause implications for metabolism and composition of the whole cells [23,24,25,26,27,30,31,32,33,34,35,36,37].

In this study, the effect of vitamin C on the composition of ROS injured normal colon cells and the effect of vitamin E on the composition of gastric cancer cells are investigated. Raman spectroscopy and imaging are used for biochemical and structural characterization of normal colon cells, normal colon cells exposed to oxidative stress generated by Tert-butyl hydroperoxide (tBHP), normal colon cells exposed to oxidative stress and vitamin C treatment, as well as for gastric cells and gastric cells exposed to vitamin E treatment. The influence of different concentrations of vitamin C under oxidative stress on chemical profile of colon cells and different concentrations of vitamin E on chemical profile of gastric cells is evaluated according to Raman spectra analysis of investigated cell lines. We decided to treat colon cells with Vitamin C and gastric cells with Vitamin E because both vitamins are the most commonly known and used antioxidants. Moreover, both vitamins are essential for the human body to function properly. Unfortunately, in the course of evolution, the human organism has lost the ability to synthesize both vitamin C and vitamin E. Therefore, it is necessary to supply them with food. Data on the minimum daily intake of vitamins C and E often appear in the literature [21,48]. However, in our research, we want to demonstrate the anti-oxidant effect of vitamins in relation to cancer diseases. With the help of the conducted research, we would like to prove that a diet rich in vitamins that are free radical scavengers can have a positive effect on inhibiting or preventing cancerous changes in the digestive system. Additionally, Raman spectroscopy was used for chemical characterization of cancer and normal colon and stomach tissues.

## 2. Results

In performed spectroscopic analysis, Raman spectroscopy and imaging were used for biochemical characterization of CCD-18 Co cells, CCD-18 Co cells treated with tBHP and different vitamin C concentrations, HTB-135 cells, HTB-135 cells treated with different vitamin E concentrations (but the same concentrations as for vitamin C), and Caco-2 and CRL-7869 cells as reference cell samples of cancerous colon cell line and normal stomach cell line, respectively.

In general, Raman vibrational spectra consists of two interesting regions, the Raman fingerprint region of 500–1800 cm^−1^ and the high frequency region of 2700–3100 cm^−1^ (the region 1800–2700 cm^−1^ is excluded from consideration through the lack of Raman bands). In this manuscript, we focused on most informative fingerprint region of 500–1800 cm^−1^.

To properly rise to biochemical changes in both normal and cancerous human colon and stomach cell lines and tissues by Raman spectroscopy and imaging, we will closely inquire into the way in which the Raman-based method responds to generated ROS and changes related to vitamin supplementation. The experiments will extend our knowledge of the protective function of antioxidants and veritable influence of the ROS generation on cancer development by RS.

All Raman spectra contain bands assigned to specific chemical structures based on vibrational features of molecules within an analyzed cell. Table 1 features the main chemical constituents which can be identified based on their vibrational properties in analyzed gastrointestinal human tissues, human colon and gastric cells in normal and oxidative stress conditions generated by tBHP adding (discussed later in the manuscript), and upon vitamin C and vitamin E supplementation.

Spectroscopic analysis by Raman spectroscopy and imaging was started with gastrointestinal samples without any supplementation.

Figure 1 characterizes single, human normal colon CCD-18 Co cell by depicting the microscopic image, Raman image of single cell constructed based on Cluster Analysis (CA) method, Raman images of all clusters identified by CA assigned to nucleus (red), mitochondria (magenta), lipid-rich regions (blue, orange), cytoplasm (green), membrane (light grey), and cell environment (dark grey) and representative average spectra from fingerprint region (500–1800 cm^−1^).

The areas defined by using Cluster analysis were assigned to different cell organelles based on the analysis of average Raman spectra. For example, for the magenta color spectrum, we observed the most intense peak for 1584 cm^−1^ and also 751 cm^−1^. Both these peaks are typical for cytochrome c, that is why we assigned this color for area typical for mitochondria; for lipid-rich regions, we noticed the most intense peaks at 1444 and 1453 cm^−1^ (moreover, we observed very intense peaks at peaks at 2854 cm^−1^; high-frequency region is not presented in the manuscript). The red region was assigned to nucleus because for the average spectrum, the intense peaks at 785 and 1092 cm^−1^ typical for nucleic acids are characterized. For the cytoplasm, the peaks were less intense compared to the other regions with spectral features most typical for proteins.

Results for CCD-18 Co cells treated with tBHP, 5 µM of vitamin C upon ROS generation, and Caco-2 cells are also presented below.

Raman bands in spectra presented in Figure 1, Figure 2, Figure 3 and Figure 4 on panels D are ascribed to nucleic acids, amino acids, proteins and lipids, thereby providing chemical characterization of CCD-18 Co and Caco-2 cells (see Table 1).

One of the major goals of the undertaken research was the biochemical analysis of human normal colon cell line in physiological or oxidative stress conditions, demonstrating the antioxidant properties of vitamin C and comparative analysis of normal human colon cells in different conditions with human cancerous colon cells Caco-2 based on the vibrational features by using label-free Raman spectroscopy and imaging. Figure 3 shows Raman spectra and imaging for cancerous human colon cells Caco-2.

One can see from Figure 1, Figure 2 and Figure 3 that, using Raman spectroscopy, it is possible to obtain well-resolved vibrational spectra to characterize the biochemistry of single cells.

Figure 4 presents microscopy. Raman data obtained for CCD-18 Co human normal colon cell treated by using 50 µM tBHP and upon vitamin C supplementation in exemplary concentration equals to 5 µM. Remaining variants of the tested samples supplemented with 50 µM tBHP concentration and various concentrations of vitamin C (25 µM and 50 µM) are presented in Appendix A.

The same spectroscopic analysis by using Raman spectroscopy and imaging was performed for human normal and cancer gastric cells CRL-7869 and HTB-135, respectively.

Figure 5, Figure 6 and Figure 7 characterize single HTB-135 and CRL-7869 cells by depicting the microscopy images, Raman images of single cells constructed based on Cluster Analysis (CA) method, Raman images of all clusters identified by CA assigned to nucleus, mitochondria, lipid-rich regions, cytoplasm, membrane, and cell surroundings, and representative cluster spectra from fingerprint region (500–1800 cm^−1^).

Results for pure HTB-135 cancer gastric cells, gastric cells treated with 5 µM of vitamin E, and normal gastric CRL-7869 are also presented. Signals in spectra presented in Figure 5, Figure 6 and Figure 7 are ascribed to nucleic acids, amino acids and proteins and/or lipids, thereby providing biochemical characterization of HTB-135 and CRL-7869 cells (see Table 1).

Figure 5 shows Raman spectra and Raman imaging for untreated normal gastric cells CRL-7869 measured in PBS.

Figure 6 shows the results of the same type of spectroscopic analysis performed for gastric cancer cells HTB-135.

Figure 7 presents microscopy and Raman data obtained for HTB-135 human cancer gastric cells upon vitamin E supplementation in exemplary concentration equals to 5 µM. Remaining variants of the tested samples supplemented with various concentrations of vitamin E (25 µM and 50 µM) are presented in Appendix A.

For further detailed analysis, we decided to choose frequencies highlighted in Figure 1, Figure 2, Figure 3, Figure 4, Figure 5, Figure 6 and Figure 7 in blue because vibration of 1078 cm^−1^ is typical for nucleic acid, vibrations of 1258 and 1658 cm^−1^ are typical for proteins and vibration of 1444 cm^−1^ is typical for lipids. As we can see from all presented Figures, the content of nucleic acids, proteins and lipids is different in different cell structures, thus the intensities of the bands assigned to these components in Raman spectra shown in panels D are different (in principle, it is also the basis for the operation of CA algorithms), allowing the identification of nucleus (intense nucleotide vibrations at 1078 cm^−1^), mitochondria (intense cytochrome vibrations at 1582 cm^−1^), lipid rich regions (intense lipid vibrations at 1444 cm^−1^), membrane (intense lipid/protein vibrations at 1444/1258/1658 cm^−1^), cytoplasm (intense cytoskeleton vibrations at 1444/1658 cm^−1^ and 1640 cm^−1^ typical for water).

The results obtained by microscopy and spectroscopic techniques for human normal and cancer cells of colon and stomach were completed with data recorded for healthy and cancer human colon and stomach tissues.

Figure 8 shows the data obtained by Raman spectroscopy and imaging of human normal and cancer colon tissues. As for single cells, Raman spectra are characterized by peaks corresponding to nucleic acids, proteins and lipids including unsaturated fraction.

Detailed analysis of data presented in above figures with results from Raman spectral analysis shows that Raman spectra typical for human colon cells and tissues are characterize by changes in the intensity of the bands occurring at c.a. 750 cm^−1^ (nucleic acids, DNA, tryptophan, nucleoproteins), c.a. 854 cm^−1^ (phosphate groups), 1004 cm^−1^ (phenylalanine), 1085 cm^−1^ (phosphodiester groups in nucleic acids), 1126 cm^−1^ (saturated fatty acids), 1172 cm^−1^ (cytosine, guanine), 1239 cm^−1^ (nucleic acids, Amide III), 1268 cm^−1^ (Amide III (C–N stretch + N–H bend)), 1304 cm^−1^ (lipids), 1334 cm^−1^ (CH_3_CH_2_ wagging vibrations of collagen), 1444 cm^−1^ (lipids (predominantly) and proteins), 1584 cm^−1^ (phosphorylated proteins) and 1664 cm^−1^ (Amide I (C=O stretch)), confirming that chemical information about the composition of cancerous and normal human colon cells and tissues can be obtained based on vibrational features of samples.

Moreover, Raman imaging allows not only to characterize colon samples quantitatively, but also to provide information about spatial distribution of chemical compounds in analyzed specimens and single cells.

Raman imaging study and Raman cluster analysis of colon tissues have shown also higher structural heterogeneity of normal tissue in a form of 3 clusters (red—protein-rich profile, blue—lipid-rich profile, purple—mix of the lipid and protein content with the predominant role of the protein) and higher homogeneity of cancer tissue in a form of 2 clusters (red—protein-rich profile, purple—mix of the lipid and protein content with the predominant role of the protein), indicating that colon cancerous transformations cause more uniform alterations during pathological processes (the human cancer colon tissue was obtained directly from the centre of tumor mass) [51].

Figure 9 shows the data obtained for normal and cancer stomach tissues by Raman spectroscopy and imaging including Raman maps and spectra characterizing different areas of samples.

Once again, based on Raman data presented in Figure 9, we can characterize gastric human tissues by bands at 750, 854, 1004, 1085, 1126, 1172, 1239, 1268, 1304, 1334, 1444, 1584 and 1664 cm^−1^ which provide chemical information about the composition of cancerous and normal human gastric samples shown in a form of chemical images. Moreover, based on vibration features of tissues, the difference in spectral profile between cancerous and normal tissue can be detected for nucleic acids, proteins and lipids, and the structural differences in regions within tissue areas can be presented in the form of 3 clusters for normal tissue (red—protein-rich profile, blue—lipid-rich profile, purple—mix of the lipid and protein content with the predominant role of the protein) and 2 clusters for cancerous tissue (red—protein-rich profile, purple—mix of the lipid and protein content with the predominant role of the protein)**.** The main role of nucleic acids, proteins and lipids in differentiation of normal and cancer human colon and stomach tissues has been confirmed also by using chemometric method. Figure 10 shows the loading plots obtained based on Raman spectra using PCA. One can see from Figure 10 that frequencies highlighted in blue in Figure 1, Figure 2, Figure 3, Figure 4, Figure 5, Figure 6 and Figure 7 for cells are most intense in loading plots 1078, 1258, 1444, 1658 cm^−1^. Moreover, Figure 10 confirms that bands selected for further analysis are the same for human colon and stomach tissues and cells and universal meaning.

## 3. Discussion

Gastrointestinal cancer is one of the most common cancers globally, and oxidative stress is associated with the occurrence of carcinogenesis including CRC [3] and GC [47]. Plant-derived compounds can apply as antioxidants, either by acting as ROS scavengers or by stimulating intracellular antioxidant enzymes [52].

Vitamins are absorbed in the gastrointestinal tract, especially in the intestine and in the stomach, so the organs have been selected perfectly correctly. We did not want to automatically repeat identical measurements for the intestine and stomach, the more so as we focused on demonstrating the possibility of tracing the protective effect of vitamins with spectroscopic methods such as Raman imaging.

Vitamin C is a powerful antioxidant that helps to protect cells from oxidative stress. By interacting with vitamin E, it protects the circulatory system against free radicals. In addition, vitamin C helps in the regeneration of the reduced form of vitamin E. Absorption of vitamin C takes place mainly in the intestine; therefore, in our research, we took steps to analyze its effect on cells obtained from the intestine segment. Worth noting is also the fact that vitamin C absorption takes place at different stages of the food path through the digestive system. Absorption of digestive products takes place mainly in the small intestine, with the participation of the mucosa, equipped with numerous intestinal villi. Each villus contains blood and lymph vessels, due to which nutrients are absorbed into the circulatory system and delivered to the farthest corners of the body.

Moreover, vitamin E can block the production of carcinogenic nitrosamines, which are formed in the stomach from food-derived nitrites. It also protects against the development of tumors by increasing the immune function. Vitamin E is a fat-soluble antioxidant that stops the body’s production of reactive oxygen species (ROS) that occurs when fat is oxidized. Vitamin E is a fat-soluble antioxidant that stops the body’s production of reactive oxygen species (ROS) that occurs when fat is oxidized. As is well known, fat digestion takes place in the final part of the stomach, i.e., in the duodenum. For this reason, vitamin E was chosen as the antioxidant, the effect of which we investigated on the cells of the stomach.

In addition, as mentioned, both tested vitamins are very popular antioxidants, therefore the idea of this publication is to test their action. Our intention was to present the difference in the effect of selected vitamins on the digestive system and show the similarity of the working mechanisms of cancerous and healthy cells subjected to oxidative conditions.

Vitamin C is a natural antioxidant commonly available in fruits and vegetables [14] or as a supplement [12]. In our study, the effect of vitamin C on viability and composition of human colon cell CCD-18 Co exposed to oxidative stress was investigated, and final conclusions confirm the anti-oxidant properties of this compound. CCD-18 Co cells are human fibroblast cells used in many studies as model normal colon cells, and different compounds such as linalool [53], Pt/MgO nanoparticles [54], or L-buthionine sulfoximine (BSO) [55] were already reported to induce oxidative stress in CCD-18 Co cells. In our study, tBHP was used as an oxidative stress inducer, because this compound is also widely exploited to effectively induce oxidative stress in vitro [56,57].

The effect of tBHP and vitamin C on viability of CCD-18 Co cells was determined by XTT tests. Viability of CCD-18 Co culture decreased in the presence of tBHP, but addition of vitamin C greatly improved viability of CCD-18 Co cells.

Stimulatory effect of vitamin C on metabolism of CCD-18 Co cells confirms results from a previous study [50]. The cytotoxic effects of tBHP can be explained by loss of glutathione, lipid peroxidation and hemolysis. In addition, tBHP significantly increases the permeability of cell membranes and impairs the metabolic pathway leading to the synthesis of ATP and irreversible DNA damage. These effects were attributed to both radical and non-radical mechanisms [58].

Over the course of the analysis of Raman spectra recorded for normal CCD-18 Co cells and CaCo-2 cancerous cells of the human colon, based on the intensity of the bands assigned to individual biochemical components, it can be noticed that vibrational spectroscopy effectively provides information on changes in the content of individual components depending on the type of analyzed components.

Figure 11 shows the spectrum typical for the CaCo-2 cancer cell line (marked in red) and the spectrum typical for CCD-18 Co normal line (marked in blue) generated using Cluster Analysis for cells as a single cluster (Panel A) and PCA pairwise analysis of these spectra (Panel B) and spectra characteristic for cells single organelles (Panel C).

The analysis of the PCA results presented in Figure 11 as well in Figure 10 shows that the most significant differences between normal and cancerous cells for human colon are observed for the following frequencies: 1004, 1078, 1256, 1304, 1334, 1444, 1584 and 1658 cm^−1^, which, according to the literature reports, can be assigned to individual cell components such as DNA, RNA, lipids, proteins or unsaturated fatty acids [49].

Higher intensities of bands assigned to proline/hydroxyproline/tyrosine (854 cm^−1^), phenylalanine (1004 cm^−1^), Amide III (1256 cm^−1^), collagen (1334 cm^−1^) and Amide I (1658 cm^−1^) for CaCo-2 confirm higher content of proteins for cancerous cells. It was concluded that increased protein content in CaCo-2 is, inter alia, due to increased nucleic acid (RNA/DNA) content in cancer cells [45,46]. The band intensity for nucleic acid (750 cm^−1^, 1334 cm^−1^) is also higher in Raman spectral profile for CaCo-2. The band characterizing phosphorylated status of proteins (1584 cm^−1^) is also higher for cancer CaCo-2 cells than for normal CCD-18 Co cells, suggesting that the process of protein phosphorylation, involved in induction of cell proliferation, invasion, metastasis and inhibition of cell apoptosis [59], is up-regulated in cancer.

One can see from Figure 11 and Figure 12 that using Raman spectroscopy and imaging, the influence of the addition of a ROS generating agent and addition of vitamin C at various concentrations, the qualitative and quantitative analysis (discussed later) can be performed. The comparison of spectral profiles shows major differences for bands at 1004, 1078, 1258, 1444 and 1658 cm^−1^ assigned to proteins, DNA and lipids, confirming that tBHP and vitamin C treatment affect metabolism and structural composition of CCD-18 Co cells.

Analysis of Raman band intensities ratio of different bands within spectrum can provide further information regarding molecular characterization of cells. That is why, to find quantitative differences between analyzed cells types, we decided to focus on selected Raman bands typical for main chemical compounds: nucleic acid, proteins, lipids. Different ratios for selected Raman band intensities corresponding to 1004/1078 (phenylalanine/nucleic acids and phospholipids), 1004/1258 (phenylalanine/amide III), 1004/1662 (phenylalanine/amide I proteins) and 1004/1444 (phenylalanine/lipids and proteins) were determined to monitor metabolic alterations in CCD-18 Co cells including treated with tBHP or tBHP and different concentrations of vitamin C.

Differential Raman spectrum for CCD-18 Co cells subjected to tBHP (50µM), for normal cells (CTRL), (calculated based on spectra presented on Figure 1 and Figure 2) and for CCD-18 Co cells subjected to tBHP (50µM) and vitamin C (50µM) and for normal cells (CTRL) (calculated based on spectra presented on Figure 1 and Appendix A) for all clusters identified by CA is presented in Appendix A. The differential spectra for individual structures are to show even more the destructive effect of ROS and the protective effect of vitamin C in specific areas in the cell.

The comparison of obtained results including data typical for cancer CaCo-2 cells is presented in Figure 13. One must notice that during the statistical data analysis the intensity of the peak at 1004 cm^−1^ was kept constant.

One can see from Figure 13 that values of investigated ratios changed with the change in concentrations of vitamin C used in experiments.

The first investigated ratio was I_1004_/_1078_, which is a ratio of phenylalanine to phosphodiester groups in nucleic acids and phospholipids.

Value of I_1004/1078_ was definitely higher for CCD-18 Co cells than for Caco-2 cells, which confirms that the amount of nucleic acid in cancer cells is higher than in normal cells (the intensity of the peak at 1004 cm^−1^ was constant and equal to 1.0). Treatment of CCD-18 Co cells with tBHP and with different vitamin C concentrations showed slight alterations in a value of I_1004_/_1078_ compared to CCD-18 Co cells without any treatment. The higher value was observed upon supplementation with vitamin C with 5 µM concentration of ROS injured CCD-18 Co cells. It means that after generation of ROS the concentration of antioxidant is too low to protect cells from harmful effects of tBHP, and the protective effect can be seen for higher concentration of vitamin C. For the concentration of 50 µM the effect was the strongest. The lowest value for the ratio I_1004_/_1078_ was obtained for Caco-2 cancer cells. It was the expected result as the more intense peaks were observed also for cancer human colon tissue (see Figure 9I).

It has been shown that permanent changes in the genome of cells resulting from oxidative stress is the first stage characteristic of the process of mutagenesis, carcinogenesis and cell aging. The appearance of a mutation in DNA is a critical step in the process of carcinogenesis. In various types of tumors, an increase in the number of oxidative damage in DNA was observed [60]. Our results presented and discussed above support this thesis. One has to remember also that in the stage of promoting carcinogenesis, ROS can induce proliferation or apoptosis of initiating cell clones. Under the influence of ROS, the concentration of Ca^2 +^ ions increases significantly in cells, which can then activate some proto-oncogenes as well as protein kinase C and thus intensify cell proliferation and the stage of cancer promotion [61].

The next ratio of interest I_1004_/_1258_ is a ratio of the intensity of bands related to the phenylalanine to amide III (N-H and C-H bend mode). A value of I_1004_/_1258_ increased for CCD-18 Co cells treated with tBHP confirming destructive effect generated by tBHP on proteins but decreased sharply for higher vitamin C concentration (50 µM), which suggests the protective effect of this compound. The I_1004_/_1662_ ratio also corresponds to proteins, and its value is related to phenylalanine to amide I (H-bonded C=O stretch mode). A value of this ratio did not differ with statistical significance between untreated CCD-18 Co cells and Caco-2 cells, but treatment of CCD-18 Co cells with tBHP increased I_1004_/_1662_ value, with the tendency for the graduate decrease while higher vitamin C concentrations were used.

At a high concentration of ROS, and at the same time with a reduced activity of proteolytic systems, oxidized proteins accumulate in the cell. Their presence has been detected in many tissues. It has been shown that oxidative stress and protein modifications caused by ROS play a role in both the aging process and the pathogenesis of many diseases including cancer [62]. Our results obtained for ROS injured normal colon cells CCD-18 Co are consistent with these findings.

Another evaluated ratio, I_1004/1444_, comparing phenylalanine to lipids, showed that treatment of CCD-18 Co with tBHP increased a value of this ratio while higher vitamin C concentrations affected the ratio differently: further increase was observed in I_1004_/_1444_ value for 25 µM vitamin C and a substantial decrease in I_1004_/_1444_ value for 50 µM vitamin C. The observed increasing tendency confirms once again the protective effect of vitamin C against ROS, based on the third type lipids of main cells building compound: nucleic acids, proteins and lipids.

ROS can also destroy lipid structures. In general, lipid peroxidation is the chain of reactions of oxidative degradation of lipids with a final product in the form of lipid peroxides or lipid oxidation products, e.g., reactive aldehydes, such as malondialdehyde (MDA) and 4-hydroxynonenal (HNE) (the second one being known as “second messenger of free radicals”).

The detailed analysis of spectroscopic data for selected ratios (I_1004/1078_, I_1004/1444_ and I_1004/1658_) for individual cells organelles are presented in Appendix A. Moreover, PCA analysis for all individual organelles identified by CA method based on Raman spectra obtained for different culturing conditions is presented in Appendix A.

To test the classification potential for the investigated colon and stomach cell lines, normal, cancerous and supplemented by antioxidants in the form of vitamin C in different concentrations, we performed partial least squares discriminant analysis (PLSDA). PLSDA results have been presented in Appendix A.

Summarizing, analysis of Raman ratios (I_1004/1078_, I_1004/1258_ I_1004/1444_ and I_1004/1662_) shows that tBHP and different vitamin C concentration treatment altered metabolism and chemical composition of human, normal colon CCD-18 Co cells. Spectroscopic data have proved also that RS can be effectively used for ROS injury and protective role of antioxidants tracking.

Using Raman spectroscopy, the influence of the addition of a vitamin E on the vibrational spectra of the tested gastric cells was also assessed.

Figure 14 shows the average spectra for cancerous HTB-135 (red) and normal CRL-7869 (blue) human gastric cells in the fingerprint region and PCA pairwise analysis. The comparison of spectral profiles shows differences for bands at 1004, 1078, 1256, 1444 and 1658 cm^−1^ assigned to proteins, DNA and lipids. The same obtained results are consistent with data presented for human colon samples.

Under normal physiological conditions, the intracellular levels of ROS are steadily maintained to protect cells from damage. Detoxification from ROS is facilitated by non-enzymatic molecules (i.e., glutathione, flavonoids and vitamins A, C and E) or through antioxidant enzymes which specifically scavenge different kinds of ROS.

In cancer cells, high, abnormal levels of ROS can result from increased metabolic activity, mitochondrial dysfunction, peroxisome activity, increased cellular receptor signaling, oncogene activity, increased activity of oxidases, cyclooxygenases, lipoxygenases and thymidine phosphorylase, or through crosstalk with infiltrating immune cells. In mitochondria, ROS are produced as an inevitable by-product of oxidative phosphorylation. The electron transport chain encompasses complexes I-IV and ATP synthase on the mitochondrial inner membrane. Superoxide is generated at complexes I and III and released into the intermembrane space (approx. 80% of the generated superoxide) or the mitochondrial matrix (approx. 20%) and further the mitochondrial permeability transition pore in the outer membrane of the mitochondrion allows the leakage of superoxide into the cytoplasm. Growth factors and cytokines also stimulate the production of ROS to exert their diverse biological effects in cancer. Many cancers arise from sites of chronic irritation, infection, or inflammation. Recent data have expanded the concept that inflammation is a critical component of tumor progression [63,64,65]. Macrophages induce the generation of ROS within tumor cells through secretion of various stimuli, such as TNFα [66]. Oxidative stress-mediated signaling events have been reported to affect all characters of cancer cell behaviour [67]. For instance, ROS in cancer are involved in cell cycle progression and proliferation, cell survival and apoptosis, energy metabolism, cell morphology, cell–cell adhesion, cell motility, angiogenesis and maintenance of tumor stemness.

Figure 15 shows the average spectra for normal (CRL-7869) human gastric cells and cancer (HTB-135) human gastric cells exposed to different vitamin E concentrations and PCA pairwise analysis in the fingerprint region. The frequencies differentiating untreated and supplemented by using vitamin E human cancer gastric cells are also highlighted.

Ratios for selected Raman band intensities used for characterisation of CCD-18 Co cells (I_1004/1078_, I_1004/1258_, I_1004/1658_ and I_1004/1444_) were also used to monitor metabolic alterations in HTB-135 cells treated with different concentrations of vitamin E. These ratio values were also compared with other profiles: HTB-135 cells non-subjected to vitamin E exposure, and CRL-7869 cells (normal gastric cells). Values of investigated ratios as a function of vitamin E concentrations used in experiments are presented in Figure 16.

One can see from Figure 16 that the value of I_1004/1078_ (phenylalanine/nucleic acids and phospholipids) was definitely higher for CRL-7869 cells than for HTB-135 cells profiles, this observation confirms the higher amount of nucleic acid in cancer gastric cells. Treatment of HTB-135 cells with vitamin E showed the decrease in a value of I_1004_/_1078_. When smaller vitamin E concentrations (5 µM) were tested, such a relation proved that low concentration of vitamin E cannot prevent the natural ROS damage typical for cancer cells and higher concentrations of antioxidant are needed to obtain the protective effect.

The ratio I_1004_/_1258_ (phenylalanine/amide III) is typical for tracking proteins. One can see from Figure 15 that this ratio was higher for CRL-7869 cells than for HTB-135 cell profiles (the same lower amount of protein is typical for normal CRL-7869 gastric cells). The treatment of HTB-135 cells with 50 µM vitamin E forced I_1004_/_1258_ value up compared to untreated HTB-135 cells; this confirms that vitamin E modulates the expressions of proteins in analyzed cells. The higher the concentration of vitamin E, the lower the signal 1258 cm^−1^, the higher the I_1004_/_1258_ ratio. Again, an I_1004_/_1658_ (phenylalanine/amide I proteins) ratio showed a higher value for CRL-7869 cells than for HTB-135 cells, confirming the lower expression of proteins in healthy cells.

In case of an I_1004/1444_ (phenylalanine/lipids), a value of this ratio was lower for CRL-7869 cells than for untreated HTB-135 cells, but treatment of HTB-135 cells with different vitamin E concentrations (5–50 µM) decreased I_1004/1444_ values to the levels of normal gastric cell profile which may suggest the protective role of vitamin E against natural oxidative processes typical for cancer samples.

The detailed analysis of spectroscopic data for selected ratios (I_1004/1078_, I_1004/1444_ and I_1004/1658_) for individual cell organelles are presented in Appendix A. Moreover, PCA analysis for all individual organelles identified by CA method based on Raman spectra obtained for different culturing conditions is presented in Appendix A.

To test the classification potential for the investigated colon and stomach cell lines, normal, cancerous and supplemented by antioxidants in the form of vitamin E in different concentrations, we performed partial least squares discriminant analysis (PLSDA). PLSDA results are presented in Appendix A.

Summarizing, the difference in values of I_1004/1078_, I_1004/1258_, I_1004/1658_ and I_1004/1444_ found between CRL-7869 and HTB-135 cells show profound metabolic changes distinguishing cancer and normal cells.

All presented results proved that metabolic regulations in cancer may involve genetic modifications and a change in metabolite pattern, with higher amino acid level, upregulated lipogenesis and upregulation of nucleotide metabolism and synthesis [68]. Moreover, HTB-135 cells exposed to different vitamin E concentrations show further changes in I_1004/1078_, I_1004/1258_ and I_1004/1444_, indicating that vitamin E can affect intensity of Raman bands typical for nucleic acids, proteins and lipids in HTB-135 cells.

The connection between cancer risk and vitamin E has been presented in various scientific studies. The anticancer effects of vitamin E have been attributed mainly to its antioxidant, anti-inflammatory, anti-proliferative, anti-angiogenic, immune modulatory mechanisms and the inhibition of reductase enzyme—HMG CoA. Antioxidants such as vitamin E provide a protective effect through neutralization. This process occurs as a result of the donation of one of its own electrons. However, this action will not turn the antioxidants themselves into free radicals, as they are stable in both forms [69]. Vitamin E reduces the activity of free radicals, preventing the escape of electrons, and thus directly participating in the formation of peroxide [70]. The radical scavenging mechanism works by removing excess free radicals, and vitamin E is known to be one of the most abundant antioxidants, an important lipophilic antioxidant that scavenges radicals and effectively scavenges peroxide radicals [71]. Moreover, vitamin E was reported to interrupt de novo sphingolipid pathway in a prostate cancer cell line [33], and modulate DNA synthesis in erythroleukemia, prostate and breast cancer cells [35].

Moreover, increased mRNA and protein levels of PPARγ, involved in fatty acid uptake and transport, were reported in colon cancer cells (SW 480) exposed to vitamin E [72,73]. Our study shows that vitamin E influences nucleic acids, proteins and lipids in gastric cancer (HTB-135) cells.

To prove that biochemical changes can be tracked by Raman spectroscopy and imaging, we have performed also PCA analysis based on Raman spectra. Figure 17 and Figure 18 show results obtained for investigated human normal, treated or cancer cells of colon and stomach.

## 4. Materials and Methods

### 4.1. Cell Line and Cultivation Conditions

CCD-18 Co cell line (ATCC^®^ CRL-1459 ™; normal colon) used in this study was purchased from ATCC (The Global Bioresource Center). CCD-18 Co cells were cultured in ATCC-formulated Eagle’s Minimum Essential Medium with L-glutamine. For the complete growth medium, Fetal Bovine Serum was added to reach a final concentration of 10%. A fresh medium was used every 2–3 days. CaCo-2 (cancer colon) cell line was also purchased from ATCC and cultured according to the ATCC protocols. The CaCo-2 cell line was obtained from a patient—a 72-year-old Caucasian male diagnosed with colon adenocarcinoma. The biological safety of the obtained material is classified as level 1 (BSL–1). To make the medium complete, we based it on Eagle’s Minimum Essential Medium with L-glutamine, with addition of a Fetal Bovine Serum to a final concentration of 20%. The medium was renewed once or twice a week.

Hs 746T (HTB-135™; gastric cancer) and Hs 738.St/Int (CRL-7869 ™; gastric normal) lines used in this study were also purchased from ATCC (The Global Bioresource Center) and were cultured in ATCC-formulated Dulbecco’s Modified Eagle’s Medium (DMEM) (ATCC: 30-2002™) with L-glutamine, glucose, sodium pyruvate and sodium bicarbonate. For the complete growth medium, fetal bovine serum was added to reach a final concentration of 10%. A fresh medium was used every 2–3 days. All cell lines were cultivated in flat-bottom flasks made of polystyrene, possessing a cell growth surface equal to 75 cm^2^. Flasks with culture were stored in an incubator under the following conditions: 37 °C, 5% CO_2_, 95% air.

### 4.2. Chemical Compounds

L-Ascorbic acid, reagent grade, crystalline (catalogue Number: A7506-25G), Vitamin E, α-tocopherol (catalogue Number: 258024) were purchased from Merck Life Science Sp. z o.o. company. XTT proliferation Kit (catalogue Number: 20-300-1000) was purchased from Biological Industries. All chemical compounds were used without additional purification.

### 4.3. Cell Treatment with tBHP, Vitamin C and Vitamin E

CCD-18 Co and HTB-135 cells were seeded onto calcium fluoride (CaF_2_) windows (25 × 1 mm) at a density of 10^3^ cells/cm3 and incubated for 24 h.

After this time, for CCD-18 Co cells, standard growth medium was removed and 50 µM of tBHP solution—the oxidative stress agent (in grown medium)—was added simultaneously with 5, 25 or 50 µM solution of vitamin C confected in fresh culture medium; time of incubation with both supplements was 24 h. In the case of supplementation only with tBHP, the medium was removed using an aspirator and replaced with a new one with the addition of tBHP with a concentration of 50 µM.

For HTB-135 cells, 5, 25 or 50 µM of vitamin E solved in fresh culture medium was added for 24 h supplementation. For Raman measurements, the cells were washed with PBS and fixed with 4% formalin solution.

### 4.4. Tissue Sample Preparation

Colon and stomach tissue samples were collected during routine surgery as an intraoperative biological material. The non-fixed, fresh samples were used to prepare 16 µm sections. Specimens of the tissue from the tumor mass and the tissue from the safety margins outside of the tumor mass were prepared for Raman analysis by placing specimens on CaF_2_ windows. Additionally, tissue sections were stained (hematoxylin and eosin) for standard histological analysis. The histopathological analysis was performed by pathologists from Medical University of Lodz, Department of Pathology, Chair of Oncology. All procedures performed in studies involving human participants were in accordance with the ethical standards of the institutional and/or national research committee and with the 1964 Helsinki Declaration. All tissue procedures were conducted under a protocol approved by the institutional Bioethical Committee at the Medical University of Lodz, Poland (RNN/323/17/KE/17/10/2017, 17 October 2017). Written informed consent was obtained from patients.

### 4.5. Raman Spectroscopy and Imaging of Cells and Tissues

After 24 h of supplementation, cells were washed with Phosphate Buffered Saline (PBS, Gibco, catalogue Number: 10010023, pH 7.4 at 25 °C, 0.01 M) to remove any residual medium components and an excess of additives that did not penetrate inside the cells during supplementation. Furthermore, PBS was removed and cells were fixed in 4% buffered formaldehyde solution for 10 min, and rinsed again with PBS. Prepared cell samples and tissue sections on CaF_2_ windows were subjected to Raman spectroscopy measurements.

Raman spectra and images of cells were registered using the confocal microscope Alpha 300 RSA+ (WITec, Ulm, Germany) equipped with an Olympus microscope integrated with a fiber (50 µm core diameter), with a UHTS spectrometer (Ultra High Through Spectrometer) and a CCD Andor Newton DU970NUVB-353 camera operating in default mode at −60 °C in full vertical binning mode. Nikon objective lens, with 40 × magnification and a numerical aperture (NA = 1.0), were used for cell measurements carried out by immersion in PBS; for tissue sample measurements 40 × dry objective (Nikon, objective type CFI Plan Fluor C ELWD DIC-M, numerical aperture (NA) of 0.60 and a 3.6–2.8 mm working distance) was used. During the experiments, a 532 nm excitation line with the excitation laser power of 10 mW and an integration time of 0.3 s for Raman measurements for the high frequency region and 0.5 s for the low frequency region was used. All data for colon and stomach cells and colon tissues was collected and processed with the use of WITec Project Plus software. All spectroscopic and imaging data were analyzed by Cluster Analysis (CA), executed using WITec Project Plus software with Centroid model and k-means algorithm. Data normalization was performed using the normalization model by means of the Origin software.

### 4.6. Cluster Analysis

Spectroscopic data were analyzed using Cluster Analysis method. This method constitutes an exploratory form of comparative data analysis in which the observations are divided into different groups that share a set of common features—in this report it is related to vibrational characteristics. Cluster Analysis was performed using WITec Project Plus software (colon tissues, colon cells, gastric cells) or Renishaw WiRE^TM^ (stomach tissues) software with Centroid model and k-means algorithm, where each cluster is represented by one mean vector.

Cluster Analysis constructs groups (or classes or clusters) based on the principle that, within a group, the observations must be as similar as possible, while observations belonging to different groups must be different. The partition of *n* observations (*x*) into k (*k* ≤ *n*) clusters S should be done to minimize the variance (Var) according to the following formula:$$\arg\;min_{S}\sum_{i=1}^{k}\sum_{x\in Si}^{}∥x\;\mu_{i}\;{∥}^{2}=\arg\;min_{S}\sum_{i=1}^{k}\left|S_{i}\right|VarS_{i},$$
where $\mu_{i}$ is the mean of points $S_{i}$.

All Raman maps presented in this research work were constructed based on principles of Cluster Analysis described above. Number of clusters (the minimum number of clusters characterized by different average Raman spectra, which describe the variety of the inhomogeneous biological sample) was 7 and 2 or 3 for normal and cancerous cells and tissues, respectively.

### 4.7. Data Processing

Data acquisition and processing were performed using WITec Project Plus or Renishaw WiRE^TM^ software. Cosmic rays were removed from each Raman spectrum and the Savitzky–Golay method was implemented for the smoothing procedure. The background subtraction and the normalization (model: divided by norm (the spectrum divided by the dataset norm)) were performed by using Origin software. The normalization model, divided by norm, was performed according to the formula:$$V^{\prime }=\frac{V}{∥V∥}$$
$$∥V∥=\sqrt{v_{1}^{2}+v_{2}^{2}+\ldots v_{n}^{2}},$$
where $v_{n}$ is the *n*th ***V*** values.

The normalization was performed for the spectral region 500–1800 cm^−1^, which forms a peculiar fingerprint of the sample.

### 4.8. Statistical Analysis

All results regarding the analysis of the intensity of the Raman spectra as a function of the type and time supplementation are presented as the mean ± SD, *p* < 0.05, where SD—standard deviation, *p*—probability value. ANOVA analysis (Analysis of variance) was conducted using Origin software (significance level—0.05, range test—Tukey).

### 4.9. PCA Analysis

Principal component analysis (PCA) is the process of computing the principal components and using them to perform a change of basis on the data, sometimes using only the first few principal components and ignoring the rest. In data analysis, the first principal component of a set of *p* variables, presumed to be jointly normally distributed, is the derived variable formed as a linear combination of the original variables that explains the most variance. The second principal component explains the most variance in what is left once the effect of the first component is removed, and may be proceeded through *p* iterations until all the variance is explained. Summarizing, PCA is a technique for reducing the dimensionality of such datasets, increasing interpretability but at the same time minimizing information loss.

### 4.10. PLSDA

Partial least squares-discriminant analysis (PLSDA) is a versatile algorithm that can be used for predictive and descriptive modelling as well as for discriminative variable selection. PLSDA is a classification method which exhibits many similarities to a supervised application of PCA. In PCA, knowledge of membership is not required; however, in PLSDA, class membership is known and contained in categorical variables used by the algorithm. In spectroscopic applications, PLSDA aims to predict sample class membership contained in matrix **Y** based on spectroscopic data contained in matrix **X**. PLSDA utilizes analytes that have large variation in intensities across the samples and attempts to correlate them to the sample class information contained in **Y**. Other meaningful results from PLSDA are scores and loadings that describe the samples and variables, respectively. A scores plot would ideally show enough separation of the classes input as a part of the analysis, whereas a loadings plot would demonstrate the variables (i.e., spectroscopic peak features) that are significant in differentiating the sample classes. Samples have scores on each determined LV, and the spectroscopic variables have loadings for each LV. Therefore, a final result of PLSDA is the prediction of sample membership for both calibration/training sets of samples used to build the model and new samples.

The gastrointestinal tissue database included specimens from 44 patients (for each patient, normal and cancer tissues were analyzed). For each type of tissue, the average number of spectra analyzed in the experiment was over a thousand. For each type of experiment performed on the gastrointestinal cell lines, the number of cells analyzed was a minimum of 20.

PCA analysis and PLSDA analysis including Variable Importance in Projection (VIP) scores were performed using MATLAB (MathWorks, Natick, Ma, USA) with PLS-Toolbox (EigenvectorResearch Inc., Wenatchee, WA, USA).

### 4.11. XTT Cell Viability Assay

Cells were seeded at 5 × 10^3^ cells/well in a 96-well plate and incubated overnight. For experiments with CCD-18 Co cells, they were treated with 50 µM solution of tBHP—Reactive Oxygen Species generating agent and then various concentrations of vitamin C were added. After 24-h supplementation in standard cultivation conditions, cells were subjected to cell viability test with tetrazolium salts (XTT).

Colorimetric assays analyze the number of viable cells by the cleavage of tetrazolium salts added to the culture medium. This technique requires neither washing nor harvesting of cells, and the complete assay, from microculture to data analysis by an BioTek Synergy HT reader, is performed in the same microplate.

Tetrazolium salt is cleaved to formazan by a complex cellular mechanism. This bioreduction occurs in viable cells only, and is related to NAD(P)H production through glycolysis. Therefore, the amount of formazan dye formed directly correlates to the number of metabolically active cells in the culture. The intensity of the colored product is directly proportional to the number of viable cells present in the tested culture. The percentage inhibition in each assay was calculated and plotted. tBHP treatment can lead to cell DNA damage followed by cell cycle arrest.

The assay determines the metabolic activity of living cells converting tetrazolium salts to formazan, which is measured by colorimetric tests at 450 nm wavelength, from which the specific signal of the sample is obtained, and 650 nm wavelength, which is the referential sample during the test on a multi-detecting BioTek Synergy HT model reader.

The effect of tBHP and vitamin C on viability of CCD-18 Co cells was investigated. Results show (Figure 19) that tBHP can decrease viability of CCD-18 Co culture, but addition of different vitamin C concentrations can greatly improve viability of these cells.

Figure 19 shows the results of XTT test obtained for CCD-18 Co human normal colon cells supplemented with ROS generating agent at concentration 50 µM and vitamin C in various concentrations after 24 h supplementation.

The results of measurements carried out for CCD-18 Co cells with the addition of the stress compound tBHP showed that under oxidative stress conditions, the growth and development of cells with normal structure is disturbed and is initially inhibited in such a way that the cells stop at the stage they were at when stressful conditions were introduced. With prolonged exposure to unfavorable, oxidative conditions, their survival gradually begins to decline, and the weaker or less developed ones probably die already at this stage. This phenomenon is illustrated by the bar of the 24-h incubation with tBHP, for which the viability of the cells is lower than for the control sample. The attached data show that a higher concentration of the antioxidant vitamin C results in a faster increase in cell survival and proliferation when subjected to oxidative conditions. Vitamin C reduces the harmful stress effect caused by ROS. Based on XTT tests results, concentrations of vitamin C equal to 5, 25 and 50 µM were chosen for spectroscopic experiments. The same concentrations were chosen for vitamin E supplementations for experiments for gastric human cancer cells.

## 5. Conclusions

In this work, Raman spectroscopy and imaging were used to evaluate the effect of vitamin C on molecular composition of human normal colon cells (CCD-18 Co) exposed to oxidative stress (tBHP) and evaluate the effect of vitamin E on biochemical composition of gastric cancer cells (HTB-135).

Raman spectroscopy and imaging successfully differentiated various types of colon and stomach cells based on their vibrational features. Moreover, Raman spectroscopy and imaging was successfully used to visualize organelles such as nucleus, mitochondria, lipid structures, cytoplasm and cell membranes in single cells (CCD-18 Co, Caco-2, CRL-7869, HTB-135) according to vibrational bands in the fingerprint region of spectra.

Based on Raman band intensities attributed to proteins, nucleic acids and lipids, as well as ratios 1004/1078, 1004/1258, 1004/1444, 1004/1658 calculated based on them, we have confirmed the protective role of vitamin C for cells in oxidative stress conditions for label-free and non-destructive spectroscopic method. For the same ratios, we have confirmed the protective role of vitamin E for cancerous cells of human stomach, but the observed effect was weaker.

Based on calculated ratios of 1004/1078, 1004/1258, 1004/1444, 1004/1658 we have proved also that the vitamin concentration effect on biochemical composition on CCD-18 Co and HTB-135 cell can be observed.

Our results proved that low doses of vitamins C and E can reduce the risk of gastrointestinal cancer. More and greater clinical trials should be performed to define appropriate doses of vitamins in order to generate visible association between intake of antioxidants in the form of vitamin C and E and the risk of cancer of gastrointestinal tract.

Based on the obtained results as well as the statistical analysis, we concluded that Raman spectroscopy enables the detection of cancerous changes in the human colon tissues based on the identification of characteristic vibrational bands of nucleic acids, proteins and lipids, including unsaturated fatty acids.

Moreover, gastric cancer and normal tissues were analyzed by means of Raman spectroscopy and imaging, depicting the difference in the structure between cancerous and non-cancerous gastric tissue based on spectra profile with bands typical for proteins, nucleic acids and lipids in the fingerprint region.

Performed PCA analysis based on Raman spectra proved that biochemical changes in tissues and cells originating from gastrointestinal tract can be tracked successfully by Raman spectroscopy and imaging. Moreover, PCA analysis proved that this technique allows for demonstrating even the subtle differences on subcellular level of investigated biological material.

## Figures and Tables

**Figure 1 molecules-28-00137-f001:**
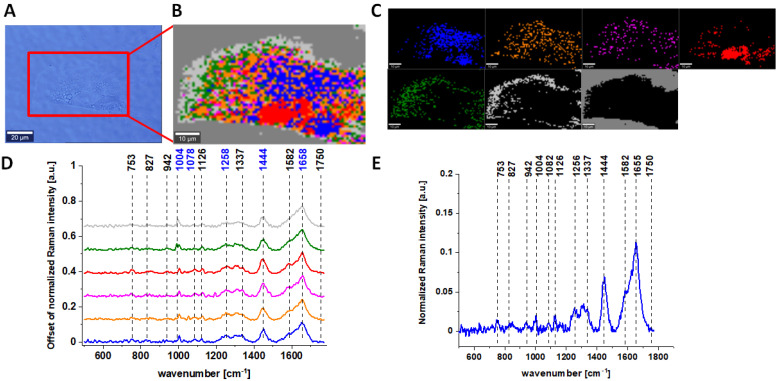
The microscopy image of normal colon CCD-18 Co cell (**A**); Raman image constructed based on Cluster Analysis (CA) method (**B**); Raman images of all clusters identified by CA assigned to nucleus (red), mitochondria (magenta), lipid-rich regions: Lipid Droplets, lysosomes, endoplasmic retikulum (blue, orange), cytoplasm (green), membrane (light grey), and cell environment (dark grey) (**C**); average Raman spectra typical for all clusters identified by CA in a fingerprint region (**D**); average Raman spectrum for the whole cell in a fingerprint region (**E**); cells measured in PBS, excitation wavelength 532 nm. Adapted with permissions [50].

**Figure 2 molecules-28-00137-f002:**
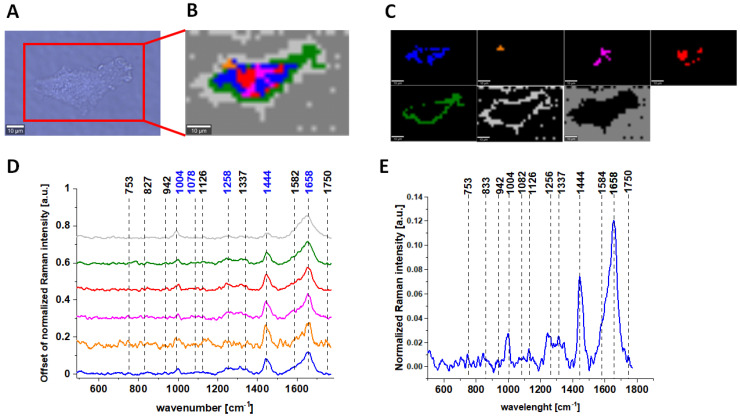
The microscopy image (**A**); Raman image constructed based on Cluster Analysis (CA) method (**B**) of normal colon cell CCD-18 Co treated with tBHP at concentration 50 µM, Raman images of all clusters identified by CA assigned to nucleus (red), mitochondria (magenta), lipid-rich regions: Lipid Droplets, lysosomes, endoplasmic retikulum (blue, orange), cytoplasm (green), membrane (light grey), and cell environment (dark grey) (**C**); average Raman spectra typical for all clusters identified by CA in a fingerprint region (**D**); average Raman spectrum for the whole cell in a fingerprint region (**E**); cells measured in PBS, excitation wavelength 532 nm.

**Figure 3 molecules-28-00137-f003:**
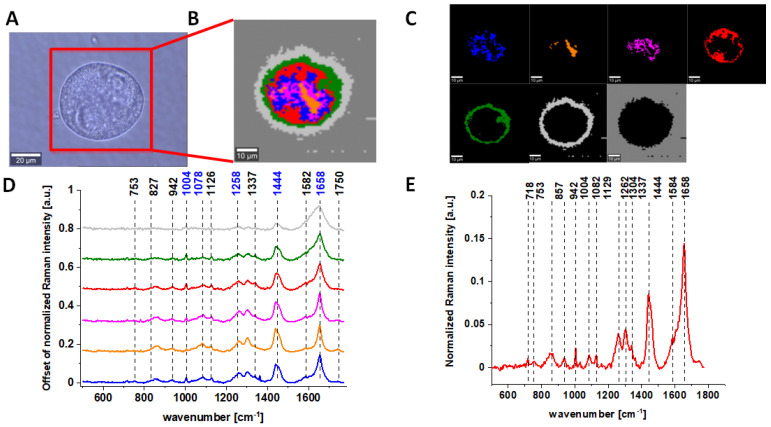
The microscopy image of colon cancer CaCo-2 cells (**A**); Raman image constructed based on Cluster Analysis (CA) method (**B**); Raman images of all clusters identified by CA assigned to nucleus (red), mitochondria (magenta), lipid-rich regions: Lipid Droplets, lysosomes, endoplasmic retikulum (blue, orange), cytoplasm (green), membrane (light grey), and cell environment (dark grey) (**C**); average Raman spectra typical for all clusters identified by CA in a fingerprint region (**D**); average Raman spectrum for the whole cell in a fingerprint region (**E**); cells measured in PBS, excitation wavelength 532 nm. Adapted with permissions [46].

**Figure 4 molecules-28-00137-f004:**
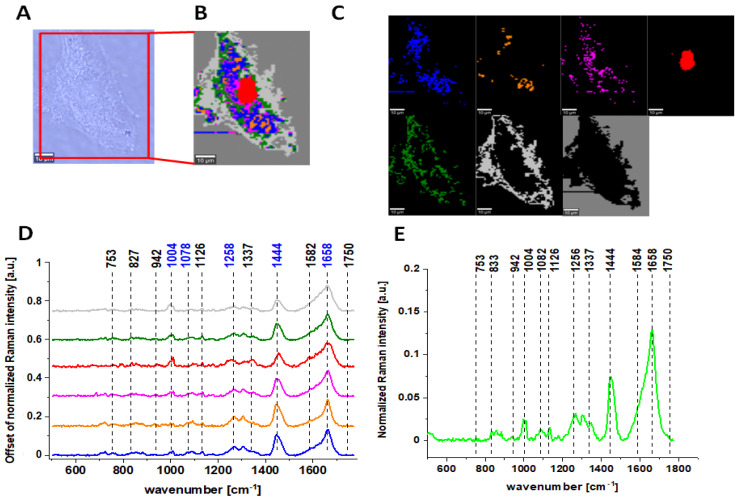
The microscopy image of normal colon CCD-18 Co cell subjected to vitamin C (5 µM) and tBHP (50 µM) treatment (**A**); Raman image constructed based on Cluster Analysis (CA) method (**B**); Raman images of all clusters identified by CA assigned to nucleus (red), mitochondria (magenta), lipid-rich regions: Lipid Droplets, lysosomes, endoplasmic retikulum (blue, orange), cytoplasm (green), membrane (light grey), and cell environment (dark grey) (**C**); average Raman spectra typical for all clusters identified by CA in a fingerprint region (**D**); average Raman spectrum for the whole cell in a fingerprint region (**E**); cells measured in PBS, excitation wavelength 532 nm.

**Figure 5 molecules-28-00137-f005:**
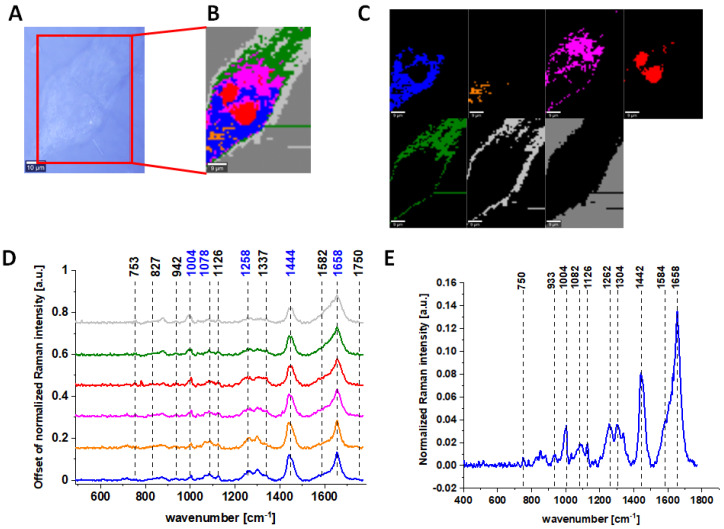
The microscopy image of human normal gastric CRL-7869 cell (**A**); Raman image constructed based on Cluster Analysis (CA) method (**B**); Raman images of all clusters identified by CA assigned to nucleus (red), mitochondria (magenta), lipid-rich regions: Lipid Droplets, lysosomes, endoplasmic retikulum (blue, orange), cytoplasm (green), membrane (light grey), and cell environment (dark grey) (**C**); average Raman spectra typical for all clusters identified by CA in a fingerprint region (**D**); average Raman spectrum for the whole cell in a fingerprint region (**E**); cells measured in PBS, excitation wavelength 532 nm.

**Figure 6 molecules-28-00137-f006:**
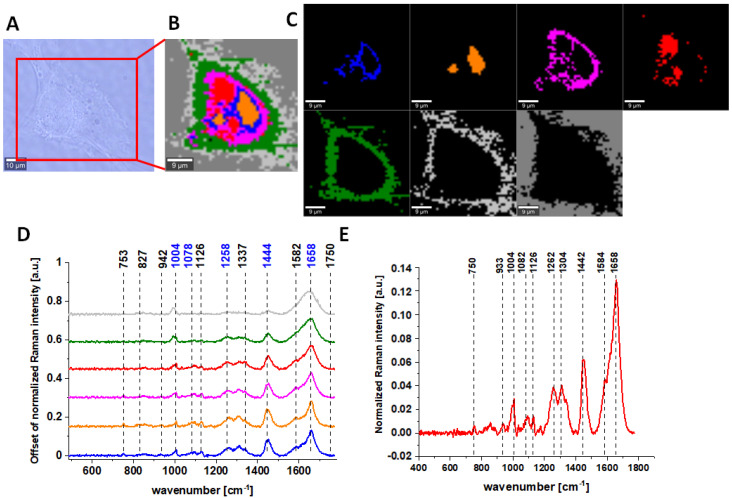
The microscopy image of human gastric cancer HTB-135 cell (**A**); Raman image constructed based on Cluster Analysis (CA) method (**B**); Raman images of all clusters identified by CA assigned to nucleus (red), mitochondria (magenta), lipid-rich regions: Lipid Droplets, lysosomes, endoplasmic retikulum (blue, orange), cytoplasm (green), membrane (light grey), and cell environment (dark grey) (**C**); average Raman spectra typical for all clusters identified by CA in a fingerprint region (**D**); average Raman spectrum for the whole cell in a fingerprint region (**E**); cells measured in PBS, excitation wavelength 532 nm.

**Figure 7 molecules-28-00137-f007:**
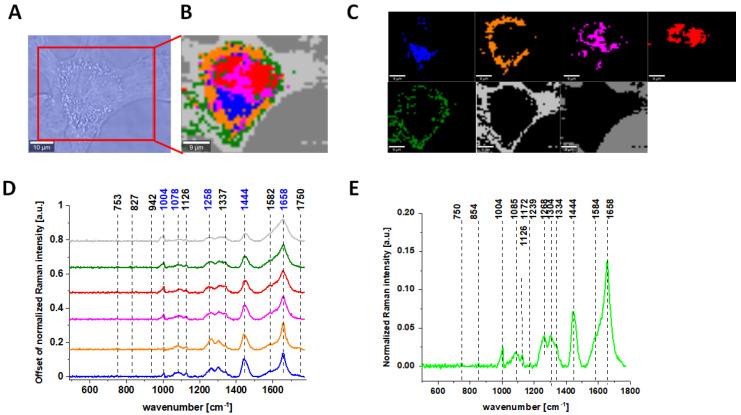
The microscopy image of human cancer gastric HTB-135 cell exposed to 5 µM vitamin E (**A**); Raman image constructed based on Cluster Analysis (CA) method (**B**); Raman images of all clusters identified by CA assigned to nucleus (red), mitochondria (magenta), lipid-rich regions: Lipid Droplets, lysosomes, endoplasmic retikulum (blue, orange), cytoplasm (green), membrane (light grey) and cell environment (dark grey) (**C**); average Raman spectra typical for all clusters identified by CA in a fingerprint region (**D**); average Raman spectrum for the whole cell in a fingerprint region (**E**); cells measured in PBS, excitation wavelength 532 nm.

**Figure 8 molecules-28-00137-f008:**
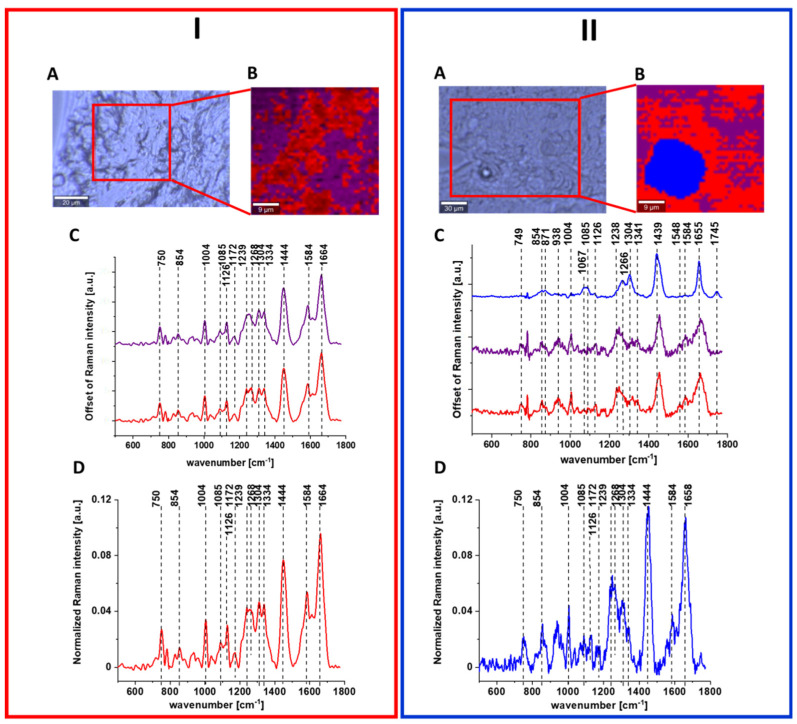
Raman image and spectral analysis of human colon cancerous tissue (**I**) and human colon normal tissue (**II**). The microscopy image (**A**), Raman image constructed by CA method (**B**), average Raman spectra typical of all clusters. Colors of the spectra correspond to colors of clusters seen in B (**C**) and average (arithmetic mean) Raman spectrum for the entire area of analyzed tissue (**D**). Excitation wavelength 532 nm. Adapted with permissions [46].

**Figure 9 molecules-28-00137-f009:**
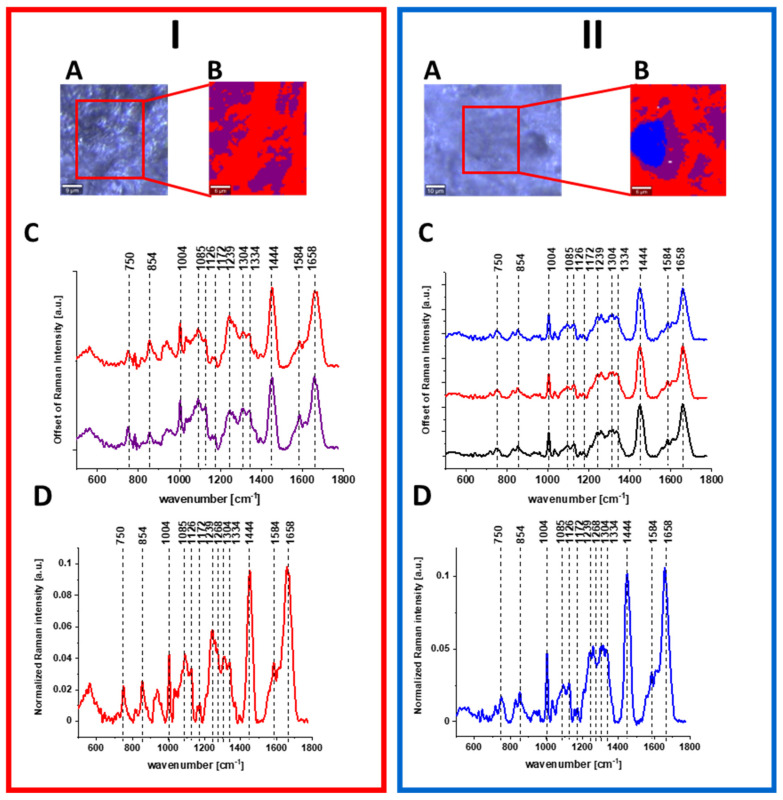
Raman image and spectral analysis of human gastric cancerous tissue (**I**) and human gastric normal tissue (**II**). The microscopy image (**A**), Raman image constructed by CA method (**B**), average Raman spectra typical of all clusters. Colors of the spectra correspond to colors of clusters seen in B (**C**) and average (arithmetic mean) Raman spectrum for the entire area of analyzed tissue (**D**) for the human stomach cancer tissue. Excitation wavelength 532 nm.

**Figure 10 molecules-28-00137-f010:**
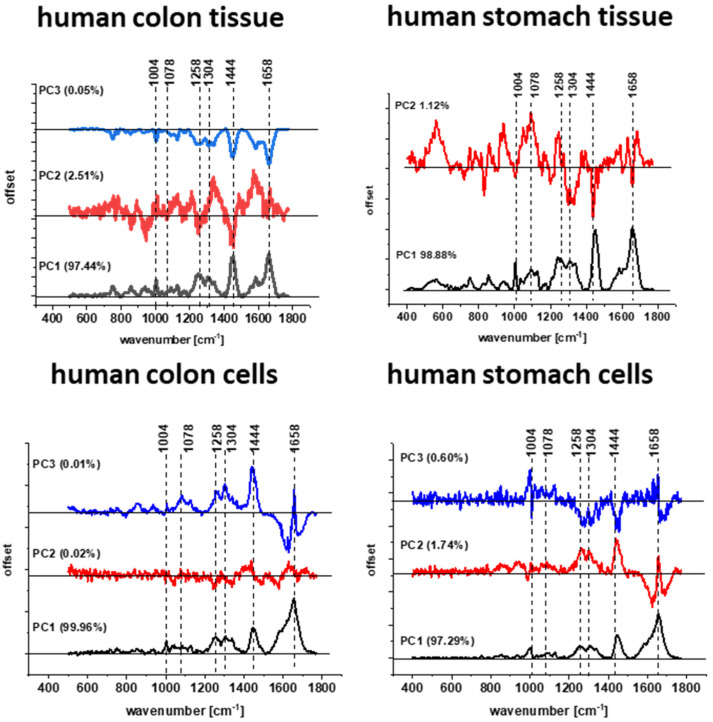
The loading plot PC1, PC2, PC3 vs. wavenumber for human colon and stomach tissues and cells obtained based on Raman spectra.

**Figure 11 molecules-28-00137-f011:**
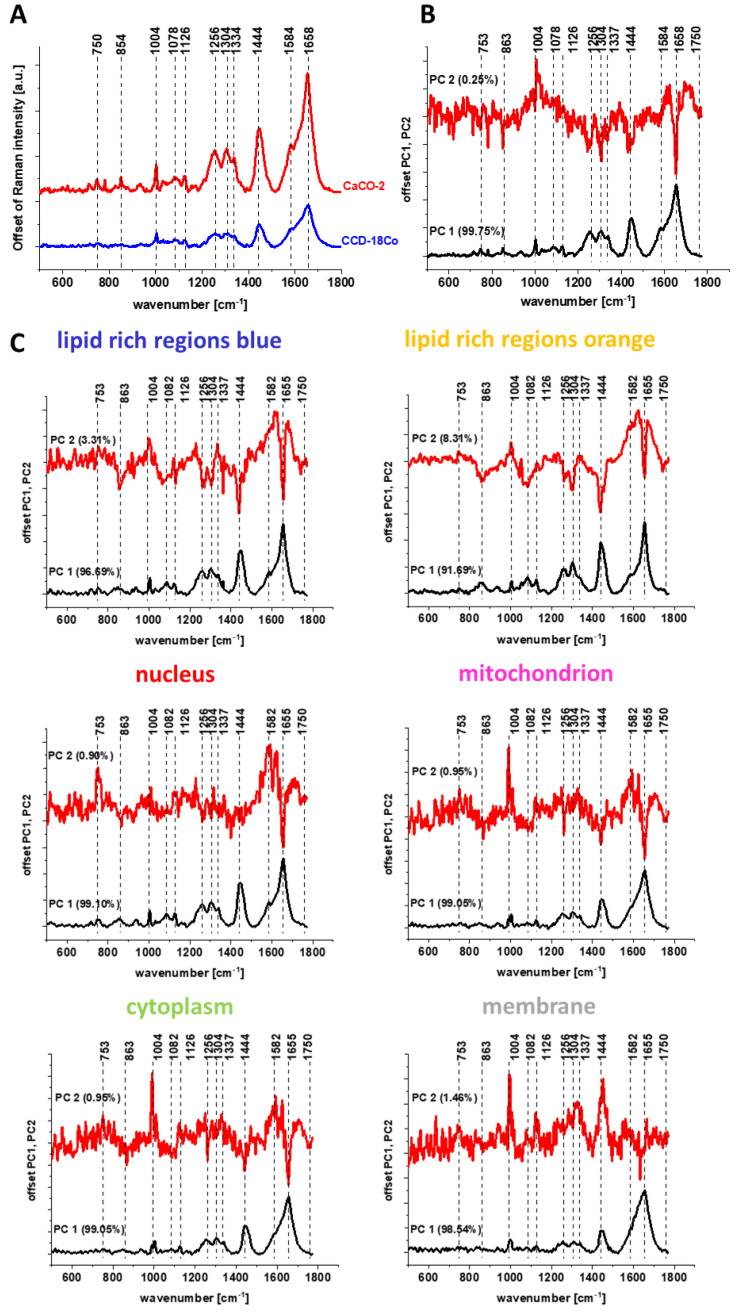
Average spectra for cancerous CaCo-2 (red) and normal CCD-18 Co (blue) human colon cells, in the fingerprint region, excitation wavelength 532 nm (Panel **A**), and PCA pairwise analysis for average spectra of CaCo-2 and CCD-18 Co cell lines, for cells as a whole (Panel **B**) and average spectra typical for single organelles such as lipid-rich regions: Lipid Droplets, lysosomes, endoplasmic retikulum (blue, orange), nucleus (red), mitochondrion (magenta), cytoplasm (green), membrane (light grey) (Panel **C**).

**Figure 12 molecules-28-00137-f012:**
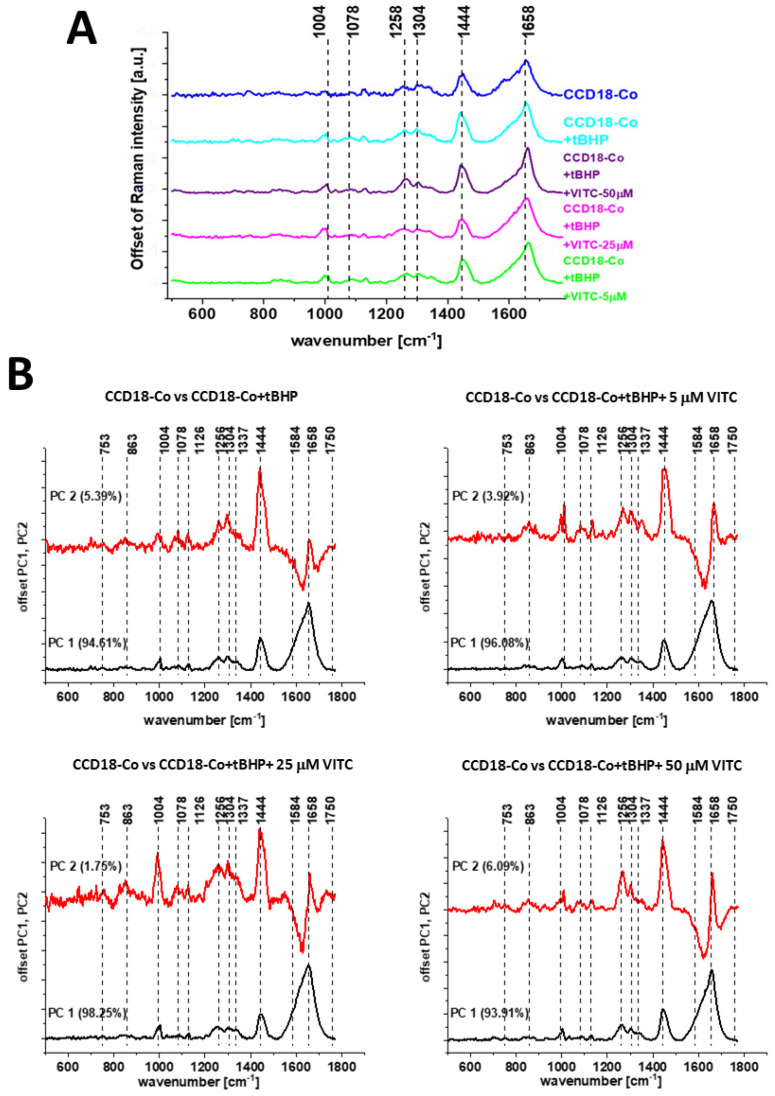
Average Raman spectra of CCD-18 Co cells, CCD-18 Co cells subjected to tBHP (50 µM) and CCD-18 Co cells subjected to tBHP upon supplementation with different vitamin C concentrations. Excitation wavelength 532 nm (Panel **A**). PCA pairwise analysis for the untreated cells and supplemented with tBHP and different concentrations of vitamin C (5, 25, 50 µM) (Panel **B**).

**Figure 13 molecules-28-00137-f013:**
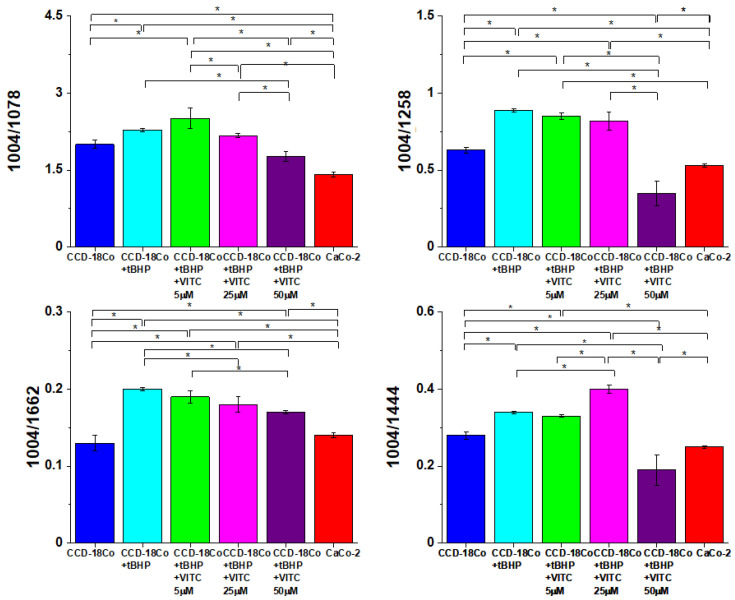
Raman band intensities ratios for selected Raman bands corresponding to 1004/1078, 1004/1258, 1004/1662, 1004/1444 in CCD-18 Co cells, in CCD-18 Co cells treated with tBHP and different concentrations of vitamin C. Plots show data obtained for 6 groups of normal human colon cells CCD18-Co: control group (labelled CCD-18-Co, blue), group supplemented with tBHP (labelled CCD-18-Co + tBHP, turquoise), group supplemented with tBHP and vitamin C in concentration of 5 µM (labelled CCD-18-Co + tBHP + VITC 5 µM, green), group supplemented with tBHP and vitamin C in concentration of 25 µM (labelled CCD-18-Co + tBHP + VITC 25 µM, magenta), group supplemented with tBHP and vitamin C in concentration of 50 µM (labelled CCD-18-Co + tBHP + VITC 50 µM, violet), group of pure colon cancer cell measured in PBS (labelled CaCo-2, red), where the statistically significant results, based on ANOVA analysis have been marked with asterisk. During the statistical data analysis, the intensity of the peak at 1004 cm^−1^ was kept constant.

**Figure 14 molecules-28-00137-f014:**
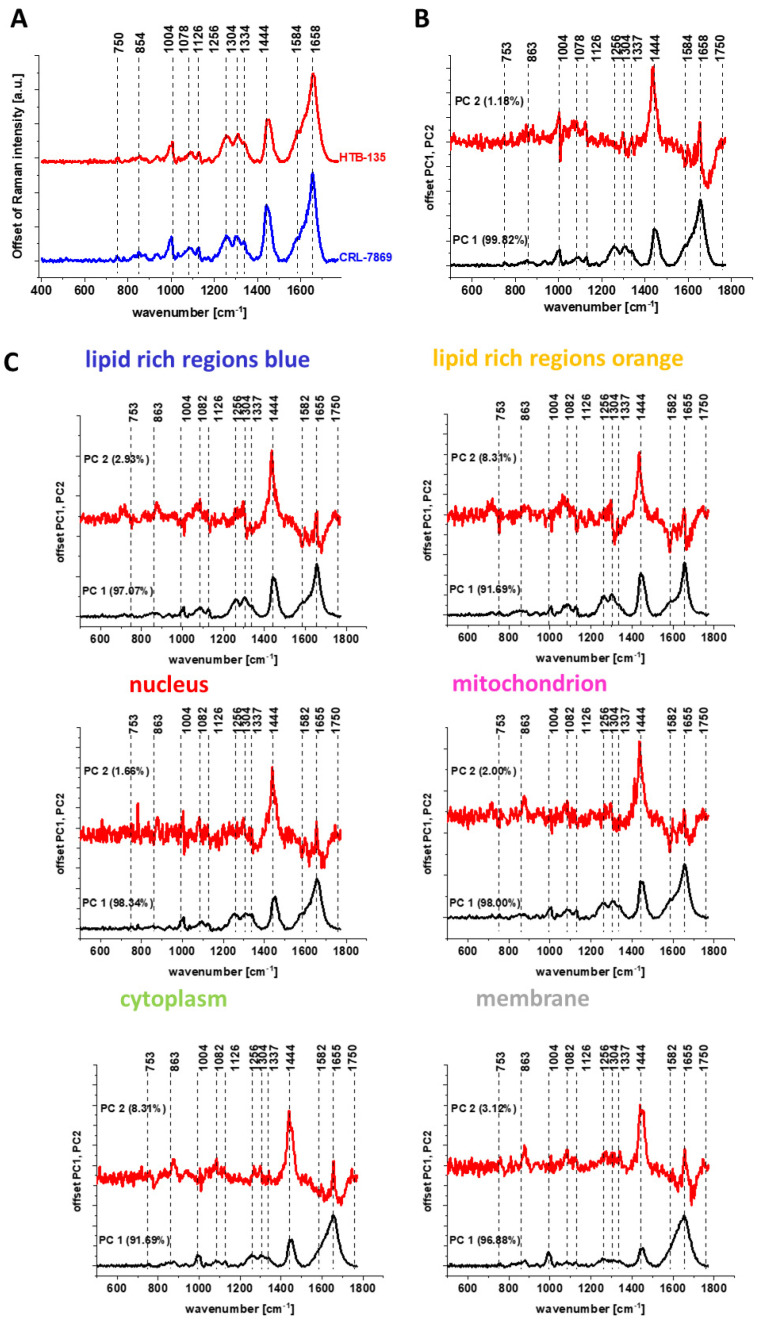
Average spectra for cancerous HTB-135 (red) and normal CRL-7869 (blue) human gastric cells, and difference spectrum (CRL-7869 minus HTB-135) in the fingerprint region, excitation wavelength 532 nm (Panel **A**) and PCA pairwise analysis for the HTB-135 and CRL-7869 cell line for a cells as a whole (Panel **B**) and average spectra typical for single organelles: lipid-rich regions: Lipid Droplets, lysosomes, endoplasmic retikulum (blue, orange), nucleus (red), mitochondrion (magenta), cytoplasm (green), membrane (light grey) (Panel **C**).

**Figure 15 molecules-28-00137-f015:**
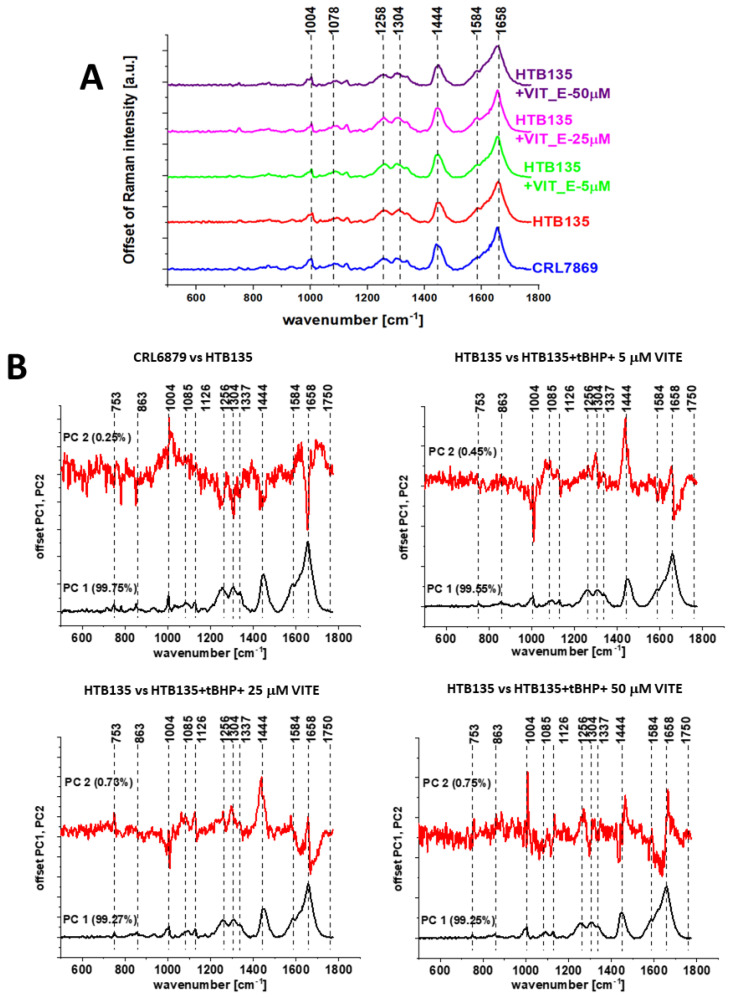
Average spectra for normal (CRL-7869) human gastric cells, cancer (HTB-135) human gastric cells, and cancer (HTB-135) human gastric cells exposed to different vitamin E concentrations in the fingerprint region. Excitation wavelength 532 nm (Panel **A**) and PCA pairwise analysis for the cells supplemented with different concentrations of vitamin E (5, 25, 50 µM) (Panel **B**).

**Figure 16 molecules-28-00137-f016:**
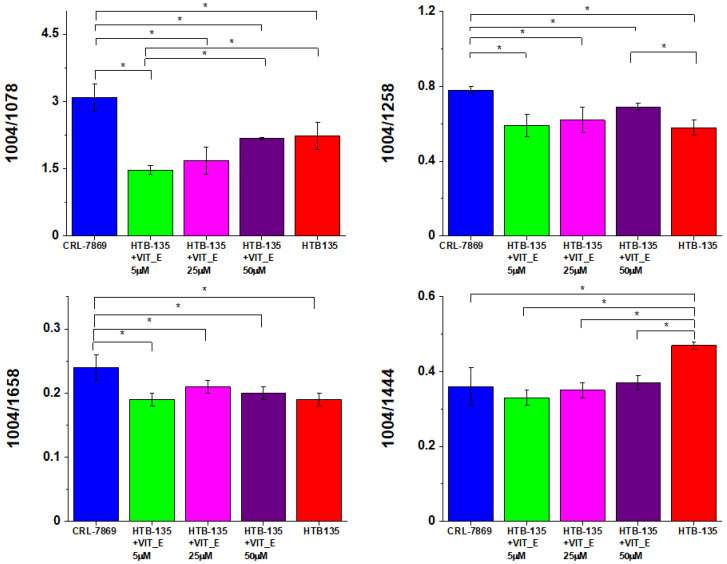
Raman band intensities ratios for selected Raman bands corresponding to 1004/1078, 1004/1258, 1004/1658 and 1004/1444, in HTB-135 cells, treated with different concentrations of vitamin E. Plots show data obtained for 5 groups of human gastric cancer cells HTB-135: control group (labelled HTB-135, red), group supplemented with vitamin E in concentration of 5 µM (labelled HTB-135 + VIT_E 5 µM, green), group supplemented with vitamin E in concentration of 25 µM (labelled HTB-135 + VIT_E 25 µM, magenta), group supplemented with vitamin E in concentration 50 µM (labelled HTB-135 + VIT_E 50 µM, violet), group of pure normal gastric cell (labelled CRL7869, blue) where the statistically significant results based on ANOVA analysis have been marked with asterisk. During the statistical data analysis, the intensity of the peak at 1004 cm^−1^ was kept constant.

**Figure 17 molecules-28-00137-f017:**
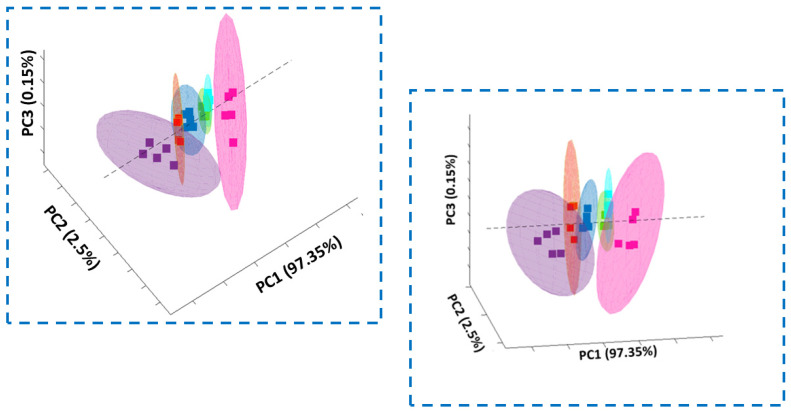
The scores 3D plot obtained from PCA (Principle Component Analysis) based on Raman spectra in two different projections for 6 groups of human colon cells: control group (blue square), normal cells treated by tBHP (turquoise square), normal cells treated by tBHP and supplemented with vitamin C in concentration of 5 μM (green square), normal cells treated by tBHP and supplemented with vitamin C in concentration of 25 μM (pink square), normal cells treated by tBHP and supplemented with vitamin C in concentration of 50 μM (violet square) and cancer cells (red square).

**Figure 18 molecules-28-00137-f018:**
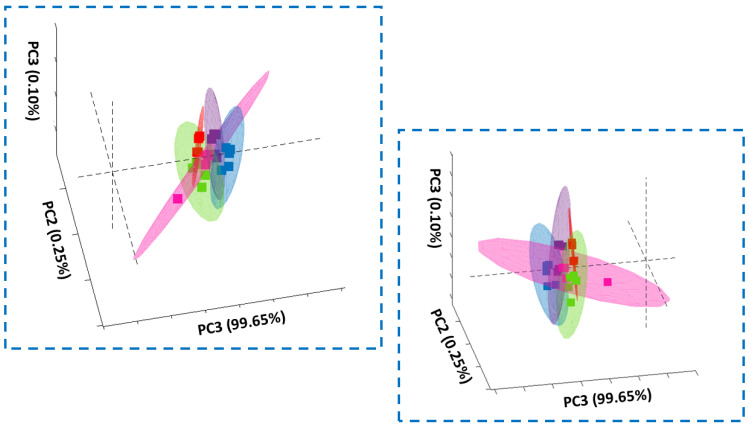
The scores 3D plot obtained from PCA (Principle Component Analysis) based on Raman spectra in two different projections for 5 groups of human stomach cells: control group (blue square), cancer cells supplemented with vitamin E in concentration of 5 μM (green square), cancer cells supplemented with vitamin E in concentration of 25 μM (pink square), cancer cells supplemented with vitamin E in concentration of 50 μM (violet square) and cancer cells (red square).

**Figure 19 molecules-28-00137-f019:**
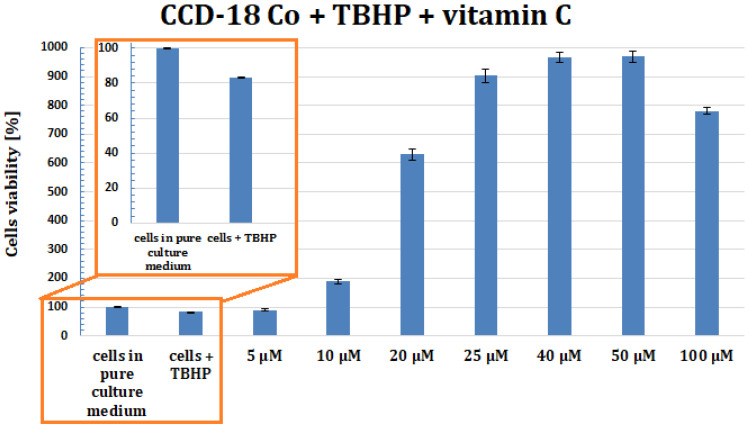
Results of XTT comparison of the percent viability for CCD-18 Co human normal colon cells treated with tBHP and supplemented with different concentrations of vitamin C after 24 h supplementation, mean ± SD, SD—standard deviation.

**Table 1 molecules-28-00137-t001:** The tentative assignments of Raman peaks for chemical characterization of CCD-18 Co (normal colon), Caco-2 (cancer colon), CRL-7869 (normal gastric), and HTB-135 (cancer gastric) cells used in this study [49].

Wavenumber [cm^−1^]	Tentative Assignments
716–723	DNA (nucleotides)
748–757	Tryptophan (protein assignment)
781–787	DNA (nucleotides)
852–858	Proline, hydroxyproline, tyrosine
992–1010	Phenylalanine
1070–1093	Phosphodiester groups in nucleic acids and phospholipids
1127–1133	Fatty acids, lipids
1256–1268	Amide III band in proteins
1299–1305	Fatty acids, lipids, phospholipids
1337–1348	Proteins, nucleic acids
1444–1447	Lipids and proteins
1582–1854	Phosphorylated proteins
1658–1664	Proteins (amide I), lipids, nucleic acids

## Data Availability

The raw data underlying the results presented in the study are available from Lodz University of Technology Institutional Data Access for researchers who meet the criteria for access to confidential data. The data contain potentially sensitive information. Request for access to those data should be addressed to the Head of Laboratory of Laser Molecular Spectroscopy, Institute of Applied Radiation Chemistry, Lodz University of Technology. Data requests might be sent by email to the secretary of the Institute of Applied Radiation Chemistry: mitr@mitr.p.lodz.pl.

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
