# Peer review of "Raman Spectroscopy and Imaging Studies of Human Digestive Tract Cells and Tissues—Impact of Vitamin C and E Supplementation"

_molecules, 2022, doi:10.3390/molecules28010137_

Round 1

Reviewer 1 Report

The paper: “Raman spectroscopy and imaging studies of human digestive tract cells and tissues – impact of vitamin C and E supplementation” proposes the use of Raman spectroscopy to characterize normal and cancer cells representing gastric and colon traits, as well as human colon and gastric samples, normal and cancerous. The organization of the work and data presentation and discussion is disordered and hard to be comprehended, and a clear work plan would be necessary to help the reader to follow the work. Within the many faults affecting the paper, some of the most relevant are detailed below.

The Title is unfitting with the text. A big amount of data on Raman spectral characterization of cells and tissues are given rather than a clear report on the “…impact of vitamin C and E supplementation”, and are more consistent with the sentence at lines 17-18: “The main goal of this study is to show the differences between healthy and cancerous tissues from the human digestive tract and human normal and cancer colon and gastric cell lines”.

In general, too numerous Figures are given in the paper, some of them also in Discussion.  The text should be thus reorganized, separating Results from Discussion, keeping only selected figures as typical examples of the working procedure, and figures summarizing the results obtained, and moving redundant Figures to Supplementary files.

Material and Methods –Cell lines used: text at lines 380-382 should be used to complete information of the cell lines used at: “2.1. Cell line and cultivation conditions”. Also, it should be clearly stated the rationale for the use of the BHP, vitamin C and vitamin E treatments on the specific cell lines.

Lines 387-389 – “Raman images of all clusters identified by CA assigned to: nucleus, mitochondria, lipid-rich regions, cytoplasm, membrane, and cell environment…”  and all Figures and text of the Results where organelles are “assigned” by “cluster CA identification” are to be revised considering that, along the text of Ref 46 (inside the Legend to Figure 13) it is clearly reported that: “CA allows for grouping a set of objects (vibrational spectra in our studies) that are similar to each other (in vibrational features in our case)”. This concept is rather repeated in this work at “2.6. Cluster analysis”, along with the presentation given in Table 1 on wavenumbers and tentative of assignment to biochemical compounds. Against this, the legends to Figures on Raman analysis of cells and tissues report on “…CA assigned to: nucleus (red), mitochondria (magenta), lipid-rich regions (blue, orange), cytoplasm (green), membrane (light grey), and cell environment (dark grey) …” So, apart the lipid rich region, indicate organelles instead of biochemical compounds.  Nuclei can be clearly observable in some cells shown in the numerous figures, but it is not substantiated how defined regions match with mitochondria. In some cases, vesicles can be observed, which could be lysosomes. Specific labeling would have to be performed to verify this, or limit the results to the biochemical components identified by Raman spectroscopy.

Minor remarks

In the Abstract, lines 14-15: the concept of the sentence “Raman spectroscopy allows to precisely characterized cell substructures (nucleus, 14 mitochondria, cytoplasm, cell membrane) and their components (proteins, lipids, nucleic acids).”should be reversed as: “Raman spectroscopy allows to precisely characterize components (proteins, lipids, nucleic acids) ...)”

Line 429 - Figure 1 is Figure 4

Figure 13, the type of cells should be repeated at the convenience of the reader.

Author Response

Reviewer 1

Open Review

English language and style

 ( ) Extensive editing of English language and style required
(x) Moderate English changes required
( ) English language and style are fine/minor spell check required
( ) I don't feel qualified to judge about the English language and style

Comments and Suggestions for Authors

The paper: “Raman spectroscopy and imaging studies of human digestive tract cells and tissues – impact of vitamin C and E supplementation” proposes the use of Raman spectroscopy to characterize normal and cancer cells representing gastric and colon traits, as well as human colon and gastric samples, normal and cancerous. The organization of the work and data presentation and discussion is disordered and hard to be comprehended, and a clear work plan would be necessary to help the reader to follow the work. Within the many faults affecting the paper, some of the most relevant are detailed below.

The Title is unfitting with the text. A big amount of data on Raman spectral characterization of cells and tissues are given rather than a clear report on the “…impact of vitamin C and E supplementation”, and are more consistent with the sentence at lines 17-18: “The main goal of this study is to show the differences between healthy and cancerous tissues from the human digestive tract and human normal and cancer colon and gastric cell lines”.

REPLY: We prefer to keep the original form of the title, because a large group of results are data for cells after supplementation. As commented by the reviewer, we focus on comparing normal and cancer cells, but we also present results for supplemented cells. Moreover results as a function of vitamin concentration are presented. For supplementation with vitamin C and E, it is a function of 3 concentrations of 5 µM, 25 µM, 50 µM.

In general, too numerous Figures are given in the paper, some of them also in Discussion.  The text should be thus reorganized, separating Results from Discussion, keeping only selected figures as typical examples of the working procedure, and figures summarizing the results obtained, and moving redundant Figures to Supplementary files.

REPLY: As suggested by reviewer the redundant Figures obtained using Raman spectroscopy has been moved to Supplementary Materials (only one type for each supplementation as an example is presented). As suggested by the Reviewer separating Results and Discussion section have been implemented in our manuscript.

Material and Methods –Cell lines used: text at lines 380-382 should be used to complete information of the cell lines used at: “2.1. Cell line and cultivation conditions”. Also, it should be clearly stated the rationale for the use of the BHP, vitamin C and vitamin E treatments on the specific cell lines.

REPLY: The description of the cluster colors code is included in the main text of the manuscript. Information on the description of the colors is included in the lines number 355-356. Moreover, a description of the colors of the clusters is included under each Figure which presents results obtained by Raman spectroscopy. Moreover, the color scheme of the individual cell substructures is always the same. Therefore, we would prefer not to add it in the "2.1. Cell line and cultivation conditions" as this paragraph deals with the conditions of cell culturing process, not the description of the results obtained for them.

Vitamins are absorbed in the gastrointestinal tract, especially in the intestine and in the stomach, so the organs have been selected perfectly correctly. We did not want to automatically repeat identical measurements for the intestine and stomach, the more so as we focused on demonstrating the possibility of tracing the protective effect of vitamins with spectroscopic methods; Raman imaging.

Vitamin C is a powerful antioxidant and helps to protect cells from oxidative stress. By interacting with vitamin E, it protects the circulatory system against free radicals. In addition, vitamin C helps in the regeneration of the reduced form of vitamin E.

Absorption of vitamin C takes place mainly in the intestine, therefore, in our research we took steps to analyze its effect on cells obtained from the intestine segment. Worth noting is also the fact that vitamin C absorption takes place at different stages of the food path through the digestive system. Absorption of digestive products takes place mainly in the small intestine, with the participation of the mucosa, equipped with numerous intestinal villi. Each villi contains blood and lymph vessels, thanks to which nutrients are absorbed into the circulatory system and delivered to the farthest corners of the body.

Vitamin E can block the production of carcinogenic nitrosamines, which are formed in the stomach from food-derived nitrites. It also protects against the development of tumors by increasing the immune function. Vitamin E is a fat-soluble antioxidant that stops the body's production of reactive oxygen species (ROS) that occurs when fat is oxidized. Vitamin E is a fat-soluble antioxidant that stops the body's production of reactive oxygen species (ROS) that occurs when fat is oxidized. As is well known, fat digestion takes place in the final part of the stomach, i.e. in the duodenum. For this reason, vitamin E was chosen as the antioxidant, the effect of which we investigated on the cells of the stomach.

Besides, as mentioned in the manuscript, both tested vitamins are very popular antioxidants, therefore the idea of this publication is to test their action. Our intention was to present the difference in the effect of selected vitamins on the digestive system and how similar are the mechanisms of work of cancerous and healthy cells subjected to oxidative conditions.

We decided to treat colon cells with Vitamin C and gastric cells with Vitamin E because both vitamins are the most commonly known and used antioxidants. Moreover, both vitamins are essential for the human body to function properly. Unfortunately, in the course of evolution, the human organism has lost the ability to synthesize both vitamin C and vitamin E. Therefore, it is necessary to supply them with food.

We decided to compare normal and cancer cells because both endogenous and exogenous sources of reactive oxygen species result in increased oxidative stress in the cell. Excess reactive oxygen fumed can result in damage to and modification of cellular macromolecules most importantly genomic DNA that can produce mutations. Oxidative stress modulates gene expression of downstream targets involved in DNA repair, cell proliferation and antioxidants. The modulation of gene expression by oxidative stress occurs in part through activation or inhibition of transcription factors and second messengers. The role of single nuclear polymorphism for oxidative DNA repair and enzymatic antioxidants is important in determining the potential human cancer risk. It is known that many of the suspected risk factors for cancer such as e.g. age are associated with higher levels of reactive oxygen species (ROS) and/or decreased antioxidant capabilities. Oxidative stress, an imbalance between ROS and antioxidant capacity, leads to described above damaged DNA and mutations. The main differences between normal and oxidative stressed cells (by adding tBHP or in cancer cells) are manifested in intensities of main building components of human cells: nucleic acids, proteins and lipids.

Due to the key role of reactive oxygen species in stimulating apoptosis, antioxidants can inhibit this protective mechanism because they lower the concentration of ROS. The work by Salganik presents studies which confirm that apoptosis induced in human breast cancer cells is accompanied by an increase in the production of ROS [1]. Providing the body with exogenous antioxidants such as vitamin E and vitamin C can protect against cancer and other diseases in people with congenital or acquired high levels of ROS [2-6].

The motivation for choosing supplementation with this vitamins and the scheme of the entire experiment was also presented and explained in the Introduction and in the Discussion section.

[1] Salganik RI. The Benefits and Hazards of Antioxidants: Controlling Apoptosis and Other Protective Mechanisms in Cancer Patients and the Human Population. J Am Coll Nutr. 2001;20(sup5):464S-472S. doi:10.1080/07315724.2001.10719185

[2] Brown L. A., Harris F. L., Jones D. P., Ascorbate deficiency and oxidative stress in the alveolar type II cell, Am J Physiol. 1997 Oct;273(4):L782-8. doi: 10.1152/ajplung.1997.273.4.L782. PMID: 9357853.

[3] Rahal A., Kumar A., Singh V., Yadav B., Tiwari R., Chakraborty S., Dhama K. Oxidative stress, prooxidants, and antioxidants: the interplay. Biomed Res Int. 2014;2014:761264. doi: 10.1155/2014/761264. Epub 2014 Jan 23. PMID: 24587990; PMCID: PMC3920909.

[4] Villagran M, Ferreira J, Martorell M, Mardones L. The Role of Vitamin C in Cancer Prevention and Therapy: A Literature Review. Antioxidants (Basel). 2021 Nov 26;10(12):1894. doi: 10.3390/antiox10121894. PMID: 34942996; PMCID: PMC8750500.

[5] Raymond YC, Glenda CS, Meng LK. Effects of High Doses of Vitamin C on Cancer Patients in Singapore: Nine Cases. Integr Cancer Ther. 2016 Jun;15(2):197-204. doi: 10.1177/1534735415622010. Epub 2015 Dec 17. PMID: 26679971; PMCID: PMC5736057.

[6] Osawa T, Kato Y. Protective role of antioxidative food factors in oxidative stress caused by hyperglycemia. In: Annals of the New York Academy of Sciences. Vol 1043. New York Academy of Sciences; 2005:440-451. doi:10.1196/annals.1333.050

Lines 387-389 – “Raman images of all clusters identified by CA assigned to: nucleus, mitochondria, lipid-rich regions, cytoplasm, membrane, and cell environment…”  and all Figures and text of the Results where organelles are “assigned” by “cluster CA identification” are to be revised considering that, along the text of Ref 46 (inside the Legend to Figure 13) it is clearly reported that: “CA allows for grouping a set of objects (vibrational spectra in our studies) that are similar to each other (in vibrational features in our case)”. This concept is rather repeated in this work at “2.6. Cluster analysis”, along with the presentation given in Table 1 on wavenumbers and tentative of assignment to biochemical compounds. Against this, the legends to Figures on Raman analysis of cells and tissues report on “…CA assigned to: nucleus (red), mitochondria (magenta), lipid-rich regions (blue, orange), cytoplasm (green), membrane (light grey), and cell environment (dark grey) …” So, apart the lipid rich region, indicate organelles instead of biochemical compounds.  Nuclei can be clearly observable in some cells shown in the numerous figures, but it is not substantiated how defined regions match with mitochondria. In some cases, vesicles can be observed, which could be lysosomes. Specific labeling would have to be performed to verify this, or limit the results to the biochemical components identified by Raman spectroscopy.

REPLY: Special labelling of cellular substructures - staining - was performed on lipid-rich regions in previous works from our laboratory. [1,2] In contrast, the staining of lysosomal structures was the subject of some master thesis created in our laboratory, therefore we are authorized to say that cellular substructures in this work were identified correctly. Moreover, such analyzes were published in previous works from our laboratory. [3,4,5,6,7]

[1] Brozek-Pluska, B.; Jarota, A.; Kania, R.; Abramczyk, H. Zinc Phthalocyanine Photochemistry by Raman Imaging, Fluorescence Spectroscopy and Femtosecond Spectroscopy in Normal and Cancerous Human Colon Tissues and Single Cells. Molecules 2020, 25, 2688, doi:10.3390/molecules25112688.

[2] Halina Abramczyk, Beata Brozek-Pluska, Arkadiusz Jarota, Jakub Surmacki, Anna Imiela & Monika Kopec (2020) A look into the use of Raman spectroscopy for brain and breast cancer diagnostics: linear and non-linear optics in cancer research as a gateway to tumor cell identity, Expert Review of Molecular Diagnostics, 20:1, 99-115, DOI: 10.1080/14737159.2020.1724092

[3] Imiela, A.; Surmacki, J.; Abramczyk, H. Novel strategies of Raman imaging for monitoring the therapeutic benefit of temozolomide in glioblastoma. J. Mol. Struct. 2020, 1217, 128381, doi:10.1016/J.MOLSTRUC.2020.128381.

[4] Abramczyk, H.; Surmacki, J.; Kopeć, M.; Olejnik, A.K.; Kaufman-Szymczyk, A.; Fabianowska-Majewska, K. Epigenetic changes in cancer by Raman imaging, fluorescence imaging, AFM and scanning near-field optical microscopy (SNOM). Acetylation in normal and human cancer breast cells MCF10A, MCF7 and MDA-MB-231. Analyst 2016, 141, 5646–5658, doi:10.1039/C6AN00859C.

[5] Brozek-Pluska, B. Statistics assisted analysis of Raman spectra and imaging of human colon cell lines – Label free, spectroscopic diagnostics of colorectal cancer. J. Mol. Struct. 2020, 1218, doi:10.1016/j.molstruc.2020.128524.

[6] Beton, K.; Brozek-Pluska, B. Vitamin C—Protective Role in Oxidative Stress Conditions Induced in Human Normal Colon Cells by Label-Free Raman Spectroscopy and Imaging. Int. J. Mol. Sci. 2021, Vol. 22, Page 6928 2021, 22, 6928, doi:10.3390/IJMS22136928.

[7] Brozek-Pluska, B.; Musial, J.; Kordek, R.; Abramczyk, H. Analysis of Human Colon by Raman Spectroscopy and Imaging-Elucidation of Biochemical Changes in Carcinogenesis. Int. J. Mol. Sci. 2019, Vol. 20, Page 3398 2019, 20, 3398, doi:10.3390/IJMS20143398.

Minor remarks

In the Abstract, lines 14-15: the concept of the sentence “Raman spectroscopy allows to precisely characterized cell substructures (nucleus, 14 mitochondria, cytoplasm, cell membrane) and their components (proteins, lipids, nucleic acids).”should be reversed as: “Raman spectroscopy allows to precisely characterize components (proteins, lipids, nucleic acids) ...)”

REPLY: The lines 14-15 has been reversed as it was suggested by the reviewer.

Line 429 - Figure 1 is Figure 4

REPLY: There has been a new numbering system of the figures taking into account the comments of the reviewer.

Figure 13, the type of cells should be repeated at the convenience of the reader.

REPLY: In the old numbering order in the Figure 13 we put the Raman spectra for colon normal and cancerous tissue, not for cells. In the new numbering for the colon tissue (normal and cancerous) it is Figure 9. The results for the gastric tissue (normal and cancerous) in the new numbering are presented in Figure 10. Therefore, we do not present the names of cell lines under these figures, as they are not used there.

Reviewer 2 Report

The manuscript entitled “Raman spectroscopy and imaging studies of human digestive tract cells and tissues–impact of vitamin C and E supplementation” by K. Beton and B. Brozek-Pluska represents a comparative study of normal cell lines treated subsequently with an oxidative chemical and vitamins C and E and cancer cell lines using Raman imaging and spectral analysis. The spectroscopic Raman imaging and spectral analysis were done for normal and cancerous tissues. This is comprehensive and detailed work revealed differences in spectral bands of main biomolecules, such as proteins, DNA and lipids constituting cells and issues both in normal state and after treatment with oxidative molecules and vitamins. Spectral analysis with the use of cluster analysis, principal component analysis, and partial least squares-discriminant analysis allow the authors confirming these differences. This work contains a lot of experimental information which is very important for the development next generation of diagnostic and prognostic method of cancer diseases, and would be interesting for the reader of the journal. However, some points have to be addressed.

1.       It is not clear how many repeats were performed for both viability test in Fig. 1 and Raman measurements.

2.       Fig. 1 looks a bit confusing since shows a huge increase in cell viability value up to 1000 % after treatment with vitamins. It would be beneficial to give data in a more classical view. Also there are no statistics in the first two bars.

3.       Page 3, rows 129-131, the sentences “A source of light is this technique are lasers 129 emitting light of specific wavelengths such as e.g. 355 nm (UV), 532 nm (green), 633 nm 130 (red) or 785 nm (near-infrared), encounters a molecule/sample.” should be rewritten.

4.       In figure 4 there is unassigned band between 1258 and 1337 cm-1 bands. What is the nature of this band? Please specify.

5.       Fig 13. There is no information of the color code in figure caption.

6.       The authors have investigated the differences in Raman spectra based on the intensity of the bands. However, the band intensity is sensitive to the position of Raman focus during the spectra acquisition. I am wonder how the authors follow this criteria during the study especially for the tissue Raman measurements which exhibit irregular morphology.

7.        As the follow up of the above comment. Did the authors observe any differences in the band position of the samples which can be used in the analysis?

Author Response

Reviewer 2

Open Review

English language and style

( ) Extensive editing of English language and style required
( ) Moderate English changes required
(x) English language and style are fine/minor spell check required
( ) I don't feel qualified to judge about the English language and style

Comments and Suggestions for Authors

The manuscript entitled “Raman spectroscopy and imaging studies of human digestive tract cells and tissues–impact of vitamin C and E supplementation” by K. Beton and B. Brozek-Pluska represents a comparative study of normal cell lines treated subsequently with an oxidative chemical and vitamins C and E and cancer cell lines using Raman imaging and spectral analysis. The spectroscopic Raman imaging and spectral analysis were done for normal and cancerous tissues. This is comprehensive and detailed work revealed differences in spectral bands of main biomolecules, such as proteins, DNA and lipids constituting cells and issues both in normal state and after treatment with oxidative molecules and vitamins. Spectral analysis with the use of cluster analysis, principal component analysis, and partial least squares-discriminant analysis allow the authors confirming these differences. This work contains a lot of experimental information which is very important for the development next generation of diagnostic and prognostic method of cancer diseases, and would be interesting for the reader of the journal. However, some points have to be addressed.

  1. It is not clear how many repeats were performed for both viability test in Fig. 1 and Raman measurements.

REPLY: Eight replications for the viability test were performed for each cell variant (clean, supplemented, etc.). For Raman measurements, 3 complete cell imaging was performed for each variant. However, each cell imaging consists of thousands of spectra that were collected from the selected cell. For this reason, we achieved reliable measurement statistics. The gastrointestinal tissue database includes specimens from 44 patients. For each type of tissue, the average number of spectra analyzed in the experiment was over a thousand. It is natural that in the manuscript we have provided data for selected patients. For each type of experiment performed on the cell lines the number of cells analyzed was a minimum of 20. Figure 1 has been enriched with an inset from the first two bars.

  1. Fig. 1 looks a bit confusing since shows a huge increase in cell viability value up to 1000 % after treatment with vitamins. It would be beneficial to give data in a more classical view. Also there are no statistics in the first two bars.

REPLY: The first two bars for the XTT test include the SD results, however, they are so small that they are not very visible in such a wide survival scale. The XTT test is based on a colorimetric cell test. Thanks to the dye uptake by cells, we know what activity they have. The uptake of the dye by the cells indicates active mitochondrial metabolism. And the reference point in our test are pure cells not subjected to supplementation. The calculated survival for supplemented cells was therefore converted to the reference sample and amounts to even 1000%. The more classic look of the chart would prevent the correct reading of the obtained results, which clearly show that vitamin C supplementation is an extremely effective therapy for oxidative stress.

  1. Page 3, rows 129-131, the sentences “A source of light is this technique are lasers 129 emitting light of specific wavelengths such as e.g. 355 nm (UV), 532 nm (green), 633 nm 130 (red) or 785 nm (near-infrared), encounters a molecule/sample.” should be rewritten.

REPLY: The sentence has been rewritten as it was suggested by the reviewer.

  1. In figure 4 there is unassigned band between 1258 and 1337 cm-1bands. What is the nature of this band? Please specify.

REPLY: The unmarked band in Figure 4 is the band identified as 1304 cm-1. According to the literature cited by us in the work with number [49], which we use for the spectral interpretation of the registered bands, it is a band characteristic for lipids and proteins (collagen).

[49] Movasaghi, Z.; Rehman, S.; Rehman, I.U. Raman spectroscopy of biological tissues. Appl. Spectrosc. Rev. 2007, 42, 493–541, doi:10.1080/05704920701551530.

  1. Fig 13. There is no information of the color code in figure caption.

REPLY: Figure 13 is re-numbered as Figure 9. In the tissue figures, the color code has been described in figure captions for both colon and stomach tissues. Panel I marked in red refers to cancerous tissue. In contrast, panel II marked in blue refers to normal tissue. The colors of the clusters correspond to the identified regions of the sample.

  1. The authors have investigated the differences in Raman spectra based on the intensity of the bands. However, the band intensity is sensitive to the position of Raman focus during the spectra acquisition. I am wonder how the authors follow this criteria during the study especially for the tissue Raman measurements which exhibit irregular morphology.

REPLY: We agree with you that band intensity is sensitive to the position of Raman focus during the spectra acquisition (unfortunately, our Confocal Raman microscope is not extended with True Surface mode. That's why we have fixed the Z position between three points randomly selected of the imaging interest. Moreover, all data is normalized, the normalization method is described in the text in chapter 3. Materials and methods: Data processing, Statistical analysis. The obtained data are then averaged.

  1. As the follow up of the above comment. Did the authors observe any differences in the band position of the samples which can be used in the analysis?

REPLY: The equipment was calibrated before every measurement on the silicon wafer. The analysis was performed taking into account the maximum peak position (We are aware of slightly changes in maximum band position; the resolution of the equipment is equal 3cm-1). We check the Z position for several points in the area selected for analysis and then optimize the Z positions. All data obtained in the analysis are normalized, the normalization method is described in the text. The obtained data are then averaged.

Round 2

Reviewer 1 Report

The paper: “Raman spectroscopy and imaging studies of human digestive tract cells and tissues – impact of vitamin C and E supplementation” has been revised by the Authors after the remarks of the Reviewer. However the revision is still unsatisfactorily. Some examples of unsatisfied points sufficient to reject the paper are given below:

Material and Methods –maybe the question was unclear, but now to clarify the remark was to suggest to improve the text on the cells used “(Cell line and cultivation conditions”: from line 161 in the new revised version) by adding the information provided at the heading of Table 1, to describe the kind of cells (CCD-18 Co 380 (normal colon), Caco-2 (cancer colon), CRL-7869 (normal gastric), and HTB-135 (cancer gastric) cells) in a text site more convenient for the reader.

As to the assignment of cluster colors codes to organelles, as already underlined in the previous remarks, it is to recall the concept that: “CA allows for grouping a set of objects (vibrational spectra in our studies) that are similar to each other (in vibrational features in our case)”, from Ref 46, doi:10.3390/molecules25112688. In this regard, while lipid-rich regions may be expected to be detectable for the chemical features with typical vibrational spectra, it is still difficult to recognize how other organelles can be reliably identified according to vibrational spectra. i.e., also cytoplasmic and mitochondrial membranes are containing lipids, besides proteins and so on. Looking at Raman spectra shown in the various Figures, and at Table 1, for example, wavenumbers of 1444 -1447 assigned in the table to lipids and proteins, as well as 1658-1664 assigned in the table to proteins, lipids and nucleic acids are present in almost all the colored lines “representing” organelles. And it is still to remark that bright field images are poorly helpful to identify organelles, maybe apart nuclei, and vesicle when observable could be lipid accumulations of lysosomes.

Another heavy fault, is that the text has not been duly reorganized as to the presence of data Figures and related description between Discussion and Results.

Author Response

Reviewer 1

Open Review

Comments and Suggestions for Authors

The paper: “Raman spectroscopy and imaging studies of human digestive tract cells and tissues – impact of vitamin C and E supplementation” has been revised by the Authors after the remarks of the Reviewer. However the revision is still unsatisfactorily. Some examples of unsatisfied points sufficient to reject the paper are given below:

Material and Methods –maybe the question was unclear, but now to clarify the remark was to suggest to improve the text on the cells used “(Cell line and cultivation conditions”: from line 161 in the new revised version) by adding the information provided at the heading of Table 1, to describe the kind of cells (CCD-18 Co 380 (normal colon), Caco-2 (cancer colon), CRL-7869 (normal gastric), and HTB-135 (cancer gastric) cells) in a text site more convenient for the reader.

REPLY: As suggested by the reviewer, the kind of cells has been added to the section 'Cell line and cultivation conditions'.

As to the assignment of cluster colors codes to organelles, as already underlined in the previous remarks, it is to recall the concept that: “CA allows for grouping a set of objects (vibrational spectra in our studies) that are similar to each other (in vibrational features in our case)”, from Ref 46, doi:10.3390/molecules25112688. In this regard, while lipid-rich regions may be expected to be detectable for the chemical features with typical vibrational spectra, it is still difficult to recognize how other organelles can be reliably identified according to vibrational spectra. i.e., also cytoplasmic and mitochondrial membranes are containing lipids, besides proteins and so on. Looking at Raman spectra shown in the various Figures, and at Table 1, for example, wavenumbers of 1444 -1447 assigned in the table to lipids and proteins, as well as 1658-1664 assigned in the table to proteins, lipids and nucleic acids are present in almost all the colored lines “representing” organelles. And it is still to remark that bright field images are poorly helpful to identify organelles, maybe apart nuclei, and vesicle when observable could be lipid accumulations of lysosomes.

REPLY: We agree with the reviewer that identified fatty structures. We changed the description of lipid structures to include the most possible ones, such as lipid droplets, lysosomes and endoplasmic reticulum.

In our analysis, we do not rely on bright field images and agree that they are poorly helpful to identify organelles.

It is true that some bands appear in the spectra of many organelles, but they vary greatly in intensity, that’s why we performed Raman band intensity analysis to assign clusters to subsequent organelles. Our analysis is also based on fluorescent staining.

The detection of the nucleus and lipid structures cannot raise any doubts due to the fact that, as we have shown in previous publications [1-10], we clearly linked the Raman spectral profile with these organelles using Oil Red O staining for lipids and Hoechst for the nucleus. Assignment of the appropriate cell area for mitochondria was based on the analysis of vibrations visible in the Raman spectrum assigned to cytochrome C present in the mitochondrion. The area of mitochondria was defined based on the intensity of bands 750, 1126, 1582 cm-1 which correlate with the presence of cytochrome C and constitute a fingerprint for this molecule. The fact that the above-mentioned bands clearly define the area of the mitochondrion has been proven in the works [6-9].

The cytoplasm was defined based on the Raman intensity of the cytoskeleton vibrations at 1444/1658 cm-1 and 1640 cm-1 typical for water.

An additional proof of the correctness of the analysis presented above is the profile of the Raman spectra of the analysis in the high-frequency region published in our previous work. [10]

According to the reviewer suggestion, we changed the caption of the Figures, having regard the Lipid Droplets, lysosomes and endoplasmic reticulum.

[1] Beton, K., Brożek-PÅ‚uska, B., Biochemistry and Nanomechanical Properties of Human Colon Cells upon Simvastatin, Lovastatin, and Mevastatin Supplementations: Raman Imaging and AFM Studies, 2022, Journal of Physical Chemistry B, 126(37), pp. 7088-7103.

[2] Beton, K., Wysocki, P., Brozek-Pluska, B., Mevastatin in colon cancer by spectroscopic and microscopic methods – Raman imaging and AFM studies, 2022, Spectrochimica Acta - Part A: Molecular and Biomolecular Spectroscopy 270,120726.

[3] Brozek-Pluska, B., Beton, K., Oxidative stress induced by: T BHP in human normal colon cells by label free Raman spectroscopy and imaging. The protective role of natural antioxidants in the form of β-carotene, 2021, RSC Advances, 11(27), pp. 16419-16434.

[4] Brozek-Pluska, B., Statistics assisted analysis of Raman spectra and imaging of human colon cell lines – Label free, spectroscopic diagnostics of colorectal cancer, 2020, Journal of Molecular Structure, 1218,128524.

[5] Brozek-Pluska, B., Jarota, A., Kania, R., Abramczyk, H., Zinc phthalocyanine photochemistry by raman imaging, fluorescence spectroscopy and femtosecond spectroscopy in normal and cancerous human colon tissues and single cells, 2020, Molecules 25(11),2688.

[6] Abramczyk, H., Surmacki, J.M., Brozek-Pluska, B., Kopec, M., Revision of commonly accepted warburg mechanism of cancer development: Redox-sensitive mitochondrial cytochromes in breast and brain cancers by Raman imaging, Cancers 2021,13(11),2599.

[7] Abramczyk, H.; Brozek-Pluska, B.; Kopec, M.; Surmacki, J.; BÅ‚aszczyk, M.; Radek, M. Redox Imbalance and Biochemical Changes in Cancer by Probing Redox-Sensitive Mitochondrial Cytochromes in Label-Free Visible Resonance Raman Imaging. Cancers 2021, 13, 960. https://doi.org/ 10.3390/cancers13050960

[8] Abramczyk, H., Surmacki, J.M., Brozek-Pluska, B. Redox state changes of mitochondrial cytochromes in brain and breast cancers by Raman spectroscopy and imaging, Journal of Molecular Structure, 2022, 1252,132134.

[9] H. Abramczyk, B. Brożek-PÅ‚uska, M. Kopeć, Double face of cytochrome c in cancers by Raman imaging, Sci. Rep. 12 (2022) 2120, DOI: 10.1038/s41598-022-04803-0

[10] Beton, K.; Brozek-Pluska, B. Vitamin C—Protective Role in Oxidative Stress Conditions Induced in Human Normal Colon Cells by Label-Free Raman Spectroscopy and Imaging. Int. J. Mol. Sci. 2021, Vol. 22, Page 6928, 22, 6928, doi:10.3390/IJMS22136928.

Another heavy fault, is that the text has not been duly reorganized as to the presence of data Figures and related description between Discussion and Results.

REPLY: The text between Discussion and Results has been checked and organized as it should be assigned to the presence of data Figures.

As indicated by the reviewer, some drawings have been moved from the Results section to Supplementary Materials not to confuse the reader. The Results section contains selected experimental results only. Their extensive discussion is included in the Discussion section. The manuscript is clearly divided into results and discussion parts.

Reviewer 2 Report

The authors have addressed all comments and thus improved the manuscript.

In my view the manuscript can be published in present form.

Author Response

Reviewer 2

Open Review

Comments and Suggestions for Authors

The authors have addressed all comments and thus improved the manuscript.

In my view the manuscript can be published in present form.

REPLY: Thank you very much for your response.
